# DEPT: Decoupled Embeddings for Pre-training Language Models

**Alex Iacob**[†,1,2,*]  **Lorenzo Sani**[1,2,*]  **Meghdad Kurmanji**[1]  **William F. Shen**[1,**]

**Xinchi Qiu**[1,**]  **Dongqi Cai**[1,3,**]  **Yan Gao**[1,2,**]  **Nicholas D. Lane**[1,2,**]

## Abstract

Language Model pre-training uses broad data mixtures to enhance performance across domains and languages. However, training on such heterogeneous text corpora requires extensive and expensive efforts. Since these data sources vary significantly in lexical, syntactic, and semantic aspects, they cause negative interference or the "curse of multilinguality". To address these challenges we propose a communication-efficient pre-training framework, DEPT. Our method decouples embeddings from the transformer body while simultaneously training the latter on multiple data sources without requiring a shared vocabulary. DEPT can: (1) train robustly and effectively under significant data heterogeneity, (2) minimize token embedding parameters to only what the data source vocabulary requires, while cutting communication costs in direct proportion to both the communication frequency and the reduction in parameters, (3) enhance transformer body plasticity and generalization, improving both average perplexity (up to **20%**) and downstream task performance, and (4) enable training with custom optimized vocabularies per data source. We demonstrate DEPT's potential via the first vocabulary-agnostic federated pre-training of billion-scale models, reducing communication costs by orders of magnitude and embedding memory by $4 - 5\times$.

## 1 Introduction

Language models (LMs) rely on sizable pre-training datasets to generalize across tasks (Radford et al., 2019; Brown et al., 2020), and languages (Pires et al., 2019; Artetxe et al., 2020; Zhao et al., 2024). More data boosts generalization and language acquisition (Hoffmann et al., 2022). However, scaling data creates a heterogeneous mix of **data sources**—different domains and languages—that challenges LMs. Issues like *Negative interference* (Wang et al., 2020), where diverse sources compete for capacity, and the *Curse of Multilinguality* (Conneau et al., 2020), where adding languages yields diminishing returns, especially on low-resource languages (Magueresse et al., 2020), persist.

Existing methods for pre-training on heterogeneous data are costly and complex. Multilingual models like BERT (Devlin et al., 2019), XLM (Conneau et al., 2020), and mT5 (Xue et al., 2021) require temperature-tuning of language sampling ratios for each model-tokenizer pair, involving expensive model selection to optimize perplexity (Conneau et al., 2020). Large Language Models (LLMs) such as LLaMA handle heterogeneous data with intensive "language-specific heuristics and model-based filters" (Dubey et al., 2024). However, these methods still face challenges such as vocabulary dilution (Rust et al., 2021) and sub-optimal cross-lingual/domain performance (Chang et al., 2023a).

This paper proposes a communication-efficient pre-training pipeline to address heterogeneous data challenges. Observing that custom vocabularies boost performance across languages (Rust et al., 2021) and domains (McLeish et al., 2024), we propose partially or fully *decoupling* the *embedding space* from transformer bodies. This approach optimizes embeddings for specific data sources while the transformer learns abstract representations. We introduce **D**ecoupled **E**mbeddings for **P**re-**T**raining (DEPT) in three variants, GLOB, TRIM, and SPEC (see Fig. 1), each increasingly leveraging

---

[†]Corresponding author: Alex Iacob `aai30@cam.ac.uk`; [*,**] Equal contribution; [1]University of Cambridge; [2]Flower Labs; [3]Beijing University of Posts and Telecommunications.

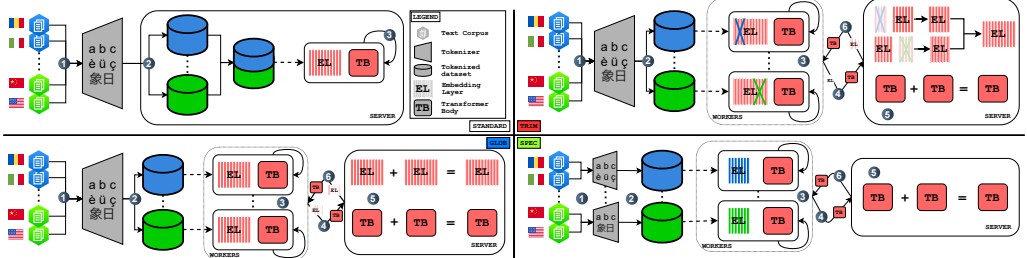

Figure 1: Pipeline for DEPT variants: TRIM (top-right), GLOB (bottom-left), SPEC (bottom-right), with the STANDARD approach (top-left). The numbered pipeline steps proceed as follows: (1) text corpora are processed into a vocabulary and tokenizer (global for STANDARD, GLOB, and TRIM; global or personalized for SPEC); (2) corpora are tokenized into a pre-tokenized dataset; (3) WORKERS train the model on their pre-tokenized data; (4) partial training results are collected; (5) results are aggregated; (6) the new model is sent to WORKERS. Steps 3–6 repeat to convergence.

specialized representations to allow pre-training with distinct domains/languages, embedding matrices, and vocabularies. For example, our SPEC variant scales the vocabulary size linearly with the number of data sources without increasing memory requirements.

DEPT enables pre-training on heterogeneous data sources with unique vocabularies and linguistic features. In the DEPT pipeline, data sources are isolated as silos, akin to clients in cross-silo Federated Learning (FL) (McMahan et al., 2017b). DEPT trains on each silo and aggregates contributions like FL clients. This work examines whether an LM can converge on data mixtures without a shared (1) output vocabulary, (2) embedding matrices, or (3) tokenization.

In summary, our work brings the following scientific contributions:

1. DEPT offers a solution to train an effective transformer body without shared global embeddings, avoiding the time, electricity, and carbon-intensive HPO tuning.
2. DEPT reduces the memory requirements of models by $\mathcal{O}((|\mathcal{V}|-\overline{|\mathcal{V}_k|})d_{\mathrm{model}})$ where $\overline{|\mathcal{V}_k|}$ is the average data source's vocabulary size, $|\mathcal{V}|$ the global vocabulary size, and $d_{\mathrm{model}}$ the embedding dimension. For multilingual models, this can save up to **80**% of the embedding-matrix size, reducing **409M** parameters for our billion-scale multilingual model.
3. DEPT-based transformer bodies show better generalization, achieving lower validation perplexities, with improvements upward of $15.3 - 20\%$ to average perplexity. DEPT models also excel in model plasticity, quickly adapting to new languages/domains. Finally, DEPT improves downstream fine-tuning performance on Natural Language Understanding tasks.
4. DEPT is communication-efficient in distributed settings, reducing communication costs compared to standard distributed data parallelism (Zhao et al., 2023) proportionally to its communication frequency. Compared to communication-efficient SGD (Stich, 2019), it obtains further reductions proportional to the size of the model embeddings. Additionally, DEPT enables vocabulary-agnostic federated pre-training for the first time.

## 2 DECOUPLED EMBEDDINGS FOR PRE-TRAINING (DEPT)

Prior work attributes the *Curse of Multilinguality* to capacity contention, vocabulary dilution (Conneau et al., 2020), and suboptimal tokenization (Rust et al., 2021). These issues affect embeddings—even though the transformer body is vocabulary-independent (Xu et al., 2024). For instance, while English may need $150\,000$ tokens (Tao et al., 2024), multilingual models allocate $250\,000$ tokens across hundreds of languages, leading to dilution, contention, and under-representation (Magueresse et al., 2020). We **propose** decoupling embeddings during training to enable custom parameters that reduce contention and vocabularies that avoid dilution and suboptimal tokenization.

We argue that training the transformer body without shared embeddings is feasible. Our **intuition** is based on evidence that: (a) transformers adapt to new languages by re-learning embeddings (Artetxe et al., 2020); (b) syntactic similarity matters more than subword sharing for performance (Pires et al., 2019); and (c) periodically re-initializing embeddings enhances plasticity (Chen et al., 2023). This suggests that transformer body performance is partly embedding-independent, allowing decoupling.

---

**Algorithm 1** Decoupled Embedding for Pre-Training (DEPT) variants: `GLOB` `TRIM` `SPEC`

---

**Require:** $S$: set of $K$ data sources, $T$: number of rounds
**Require:** $\theta_0$: initial transformer blocks, $\phi_0, \psi_0$: optional token/positional embeddings
**Require:** $\{\mathcal{D}_k\}_{k=1}^K$: source-specific datasets, $\{\mathcal{V}_k\}_{k=1}^K$: source-specific vocabularies
**Require:** `InnerOPT`: inner optimizer, `OuterOPT`: outer optimizer, e.g., AdamW and FedAvg

1: **for** each update round $t = 1, 2, \ldots, T$ **do**
2:      Randomly select a subset $S_t \subseteq S$ of data sources for round $t$
3:      **for** each data source $k \in S_t$ **in parallel do**
4:          $\theta_t^k, \phi_t^k, \psi_t^k \leftarrow \texttt{InnerOPT}(\theta_{t-1}, \phi_{t-1}, \psi_{t-1}, \mathcal{D}_k)$      ▷ `GLOB`: Global embeddings
5:          $\phi_{t-1}|_{\mathcal{V}_k} = \texttt{Trim}(\phi_{t-1}, \mathcal{V}_k)$      ▷ `TRIM`: Trim global token embeddings
6:          $\theta_t^k, \phi_t|_{\mathcal{V}_k}, \psi_t^k \leftarrow \texttt{InnerOPT}(\theta_{t-1}, \phi_{t-1}|_{\mathcal{V}_k}, \psi_{t-1}, \mathcal{D}_k)$      ▷ `TRIM`
7:          $\theta_t^k, \phi_t^k, \psi_t^k \leftarrow \texttt{InnerOPT}(\theta_{t-1}, \phi_{t-1}^k, \psi_{t-1}^k, \mathcal{D}_k)$      ▷ `SPEC`: specialized embeddings
8:          $\Delta\theta_t^k \leftarrow \theta_t^k - \theta_{t-1}$      ▷ Compute parameter update
9:          $\Delta\phi_t^k \leftarrow \phi_t^k - \phi_{t-1}$      ▷ `GLOB`: Compute global token embedding update
10:        $\Delta\phi_t|_{\mathcal{V}_k} \leftarrow \phi_t|_{\mathcal{V}_k} - \phi_{t-1}|_{\mathcal{V}_k}$      ▷ `TRIM`: Compute Trimmed embeddings update
11:        $\Delta\psi_t^k \leftarrow \psi_t^k - \psi_{t-1}$      ▷ `GLOB`+`TRIM`: global positional embedding update
12:      $\theta_t \leftarrow \texttt{OuterOPT}(\theta_{t-1}, \{\Delta\theta_t^k\}_{k \in S_t})$      ▷ Apply the updates for the transformer body
13:      $\phi_t \leftarrow \texttt{OuterOPT}(\phi_{t-1}, \{\Delta\phi_t^k\}_{k \in S_t})$      ▷ `GLOB`: Apply token updates
14:      $\phi_t \leftarrow \texttt{OuterOPT}(\phi_{t-1}, \{\Delta\phi_t|_{\mathcal{V}_k}\}_{k \in S_t})$      ▷ `TRIM`: Apply token updates
15:      $\psi_t \leftarrow \texttt{OuterOPT}(\psi_{t-1}, \{\Delta\psi_t^k\}_{k \in S_t})$      ▷ `GLOB`+`TRIM`: Apply position updates
16: **return** $\theta_T, \phi_T, \psi_T$

---

Our method, DEPT, achieves this decoupling by: (1) tokenizing data sources independently, using a global or custom vocabulary; (2) randomly initializing LM parameters; and (3) training iteratively over random source subsets (see Section 2). This contrasts with standard pre-training, which uses shared embeddings and draws random samples from a distribution of all sources.

## 2.1 METHOD

Akin to federated and meta-learning, DEPT optimizes a global parameter set $\theta$ (the transformer body) along with optional embeddings $\phi, \psi$ across data sources $S$. It trains iteratively by selecting a subset $S_t \subset S$ each round $t$. For each data source ($k \in S_t$), DEPT independently performs inner-loop optimization (`InnerOPT`, e.g., SGD) and then aggregates the transformer bodies using an outer-loop optimizer (`OuterOPT`, e.g., FedAvg). We present three variants for managing $\phi$ and $\psi$, offering progressively stronger specialization, and compare them in Section 2.4.

`GLOB` **Shared Embeddings**: Based on FedAvg-like methods, `GLOB` sends a global transformer and embeddings to each data source, which then trains locally. The updated models are aggregated via `OuterOPT`, making `GLOB` suitable for federated and centralized settings.

`TRIM` **Partially-decoupled**: Each data source gets a global transformer and embeddings but trims the token embeddings to its local vocabulary $\mathcal{V}_k$, reducing the input/output space. During `OuterOPT` aggregation, trimmed embeddings are projected to the global vocabulary.

`SPEC` **Fully-decoupled**: Each data source gets a global transformer and, when first sampled, randomly initializes specialized token/position embeddings. These remain local (never aggregated), supporting any vocabulary, including those from specialized tokenizers.

DEPT replaces the standard pre-training pipeline (Fig. 1) for broad pre-training before adaptation (Dubey et al., 2024). Algorithm 1 runs in parallel, scales with hardware, and reduces communication. Reduced communication makes it ideal for low-bandwidth settings like cross-silo FL.

## 2.2 TRIMMED EMBEDDING AGGREGATION (TRIM)

For data source $k$, trimmed embeddings $\phi_k \in \mathbb{R}^{|\mathcal{V}_k| \times d_{\text{model}}}$ are derived from global ones $\phi \in \mathbb{R}^{|\mathcal{V}| \times d_{\text{model}}}$ as $\phi_k = \mathcal{I}_k \phi$, where $|\mathcal{V}|$ is the global vocabulary size, $|\mathcal{V}_k|$ the source-specific size, and $d_{\text{model}}$ the embedding dimension. The indicator function

$\mathcal{I}_k(i,j) = \mathbb{I}[\text{the } j\text{-th token in } \mathcal{V} \text{ corresponds to the } i\text{-th local token in } \mathcal{V}_k]$ selects tokens from $\phi$. After `InnerOPT` we create $\hat{\phi}_k \in \mathbb{R}^{|\mathcal{V}| \times d_{\text{model}}}$, using zero-padding for tokens in $\mathcal{V} \setminus \mathcal{V}_k$, and use $\mathcal{I}_k^\top \in \mathbb{R}^{|\mathcal{V}| \times |\mathcal{V}_k|}$ to project $\phi_k$ back, $\hat{\phi}_k = \mathcal{I}_k^\top \phi_k$. Aggregation (`OuterOPT`) is then applied to $\{\hat{\phi}_k\}_{k \in S_t}$ with zero-padding ignored to avoid interference between tokens not shared across sources.

## 2.3 Positional Embedding Specialization (SPEC)

Unlike other variants, `SPEC` specializes both token embeddings $\phi$ and positional embeddings $\psi$, as evidence shows syntactic order-dependent properties matter more than subword sharing (Pires et al., 2019). Thus, `SPEC` is agnostic to vocabulary and sequence length, enabling federated learning without shared tokenization. Without positional specialization, `SPEC` resembles `TRIM`, but with the embedding matrix split across sources and disjoint vocabularies $\{\mathcal{V}_k\}_{k=1}^K$ such that $\mathcal{V} = \cup_{k=1}^K \mathcal{V}_k$.

## 2.4 Variant Characteristics

Table 1: Memory and communication costs of `DEPT`, where: $\mathcal{M}$ is the number of model parameters; $|\mathcal{V}|$ is the global vocabulary size; $\overline{|\mathcal{V}_k|}$ is the mean data source vocabulary size; $d_{\text{model}}$ is the embedding dimension; $N_{\text{local}} = N/T$ is the number of local steps done per iteration for a total number steps $N$; $\mathcal{L}$ is the sequence length. `GLOB` reduces comms by only communicating every $N_{\text{local}}$ steps while `TRIM` also reduces embedding size. `SPEC` brings further reductions over `TRIM` by not sharing token or position embeddings. The standard baseline is assumed to be distributed training with per-step synchronization. Concrete numbers for our models (see Table 8) are shown in Table 2.

| Method | Memory Cost | Per-step Comms Cost | Vocab Agnostic |
|--------|-------------|---------------------|----------------|
| **STD** | $\mathcal{O}(\mathcal{M})$ | $\mathcal{O}(\mathcal{M})$ | $\times$ |
| **GLOB** | $\mathcal{O}(\mathcal{M})$ | $\mathcal{O}\big(\frac{\mathcal{M}}{N_{\text{local}}}\big)$ | $\times$ |
| **TRIM** | $\mathcal{O}(\mathcal{M} - (|\mathcal{V}| - \overline{|\mathcal{V}_k|})d_{\text{model}})$ | $\mathcal{O}\big(\frac{\mathcal{M} - (|\mathcal{V}| - \overline{|\mathcal{V}_k|})d_{\text{model}}}{N_{\text{local}}}\big)$ | $\times$ |
| **SPEC** | $\mathcal{O}(\mathcal{M} - (|\mathcal{V}| - \overline{|\mathcal{V}_k|})d_{\text{model}})$ | $\mathcal{O}\big(\frac{\mathcal{M} - (|\mathcal{V}| + \mathcal{L})d_{\text{model}}}{N_{\text{local}}}\big)$ | $\checkmark$ |

In most scenarios, practitioners can deploy any of our proposals, obtaining reduced communication and memory costs as shown in Table 1. However, some settings are appropriate for a given variant.

**GLOB** resembles a standard pre-training pipeline. Although it does not explicitly decouple embeddings from the transformer, they decouple over the course of an inner-loop iteration since only local tokens influence them. As a communication-efficient form of SGD, `GLOB` reduces communication costs compared to distributed algorithms such as DDP (Li et al., 2020) or FSDP (Rajbhandari et al., 2020), which synchronize gradients at every step. However, constructing a global vocabulary requires sufficient knowledge of the dataset and may risk vocabulary dilution and capacity contention.

**TRIM** shares the same assumptions as `GLOB` and can be deployed similarly. It further reduces memory requirements for embeddings to match the data source's needs ($d_{\text{model}} \times \mathcal{V}_k$), also lowering communication costs. These savings are substantial for multilingual models with large vocabularies(Ushio et al., 2023), for instance, `mT5` and `mBART` (Xue et al., 2021; Lewis et al., 2020) allocate $40\% - 80\%$ of parameters to embeddings. Since our models use *tied weights* (Inan et al., 2017), `TRIM` restricts their output space, unlike `GLOB`, bringing a slight impact to perplexity.

**SPEC** enables pre-training across data sources without a shared vocabulary, providing `TRIM`'s benefits plus local specialization. Communication costs are minimized by transferring only the transformer body to the outer optimizer and decoupling embeddings, enabling vocabulary-agnostic training. This makes `SPEC` ideal for training a transformer body with unknown or private data. To enable inference, `SPEC` requires a global embedding matrix. While several methods exist (Section 6.1 and Appendix F), we use the straightforward approach of multi-phase adaptive pre-training (Gururangan et al., 2020), or continued pre-training with a randomly initialized matrix. This approach follows other techniques for enhancing model capabilities, e.g., long-context pre-training stages (Devlin et al., 2019; Dubey et al., 2024) and domain adaptation (Gururangan et al., 2020).

## 3 EXPERIMENTAL DESIGN

We propose DEPT as an efficient alternative to standard pre-training to address the *Curse of Multilinguality* and *Negative interference*. In this section, we conduct experiments to evaluate DEPT's performance, focusing on the following research questions:

**RQ1** Does DEPT allow us to increase the number of training tokens from heterogeneous data?
**RQ2** Does DEPT improve efficiency, in terms of memory and communication costs?
**RQ3** Does DEPT improve **zero-shot generalization** to out-of-distribution data?
**RQ4** Does DEPT improve model **plasticity** when learning new distributions?

### 3.1 EXPERIMENTAL SETUP

For our experiments, we train decoder-only transformers—currently the most relevant architectures—ranging from 125M to 1.3B parameters with 12 to 24 blocks (Tables 2 and 8). We use parameter averaging (McMahan et al., 2017a; Stich, 2019) as our OuterOpt optimizer, and AdamW (Loshchilov & Hutter, 2019) for InnerOpt. Full experimental details on our architecture, training hyperparameters (Tables 2 and 8), dataset, and baseline implementation are in Appendix A.

### 3.2 MULTI-DOMAIN AND MULTILINGUAL METHODOLOGY

To evaluate DEPT on **multi-domain** data, we use The Pile (Gao et al., 2021), which includes 22 subsets. We select 16 non-copyrighted subsets as our $K$ data sources in Algorithm 1: GitHub (GH), DeepMind Mathematics (DM), Wikipedia (WK), Common Crawl (CC), PubMed Abstracts (PA), PubMed Central (PC), USPTO Backgrounds (UB), NIH Exporter (NH), FreeLaw (FL), Enron Emails (EE), EuroParl (EP), Stack Exchange (SE), Philosophy Papers (PP), ArXiv (AX), Project Gutenberg (GU), and Hacker News (HN). Ubuntu IRC (UI) is the out-of-distribution dataset.

For **multilingual** data, we use MC4 (Xue et al., 2021) with a mix of high, medium, and low-resource languages: English (EN), Italian (IT), and Chinese (ZH) as high-resource; Serbian (SR) and Malay (MS) as medium-resource; and Swahili (SW), Urdu (UR), and Latin (LA) as low-resource. Following (Rust et al., 2021), we train unigram SentencePiece (Kudo & Richardson, 2018) tokenizers with a 50 257 vocabulary per data source. SPEC variants with optimized per-source vocabularies have the OPT suffix; otherwise, they use a global vocabulary with specialized embeddings.

### 3.3 BASELINES

We compare DEPT with standard pre-training methods from prior works (Conneau et al., 2020). General distributed SGD methods (Li et al., 2020; Rajbhandari et al., 2020), which synchronize gradients at each step and sample from all data sources simultaneously, are labeled as STD. For multilingual data, we apply temperature-weighted sampling (Devlin et al., 2019) with $\tau = 0.3$, denoted as STD ($\tau = 0.3$), as well as uniform, STD ($\tau = 0$), and proportional, STD ($\tau = 1$), sampling.[1] For multi-domain data, we use uniform and proportional sampling. Given our data sources random sampling (Algorithm 1), baselines with uniform sampling are closest to DEPT.

Additionally, we compare against the "pre-training with active forgetting" (ACT) method (Chen et al., 2023), which enhances plasticity and generalization by periodically randomly resetting embeddings. While Chen et al. (2023) transfer monolingual models between languages, we only utilize their pre-training phase due to our different settings. Like SPEC, ACT does not produce a fully trained embedding matrix and we employ the same multi-phase adaptive pre-training to create a new embedding matrix from a random initialization. Despite this similarity, SPEC is significantly more compute efficient than ACT, as it avoids extensive retraining of embeddings. Full details for how we implemented and adapted ACT can be found in Appendix A.1.3.

### 3.4 METRICS

The key characteristics for multi-domain and multilingual pre-training are model **generalization** and **plasticity**. **Generalization** refers to the model's ability to perform well on out-of-distribution

---

[1] $\tau = 0.3$ was tuned and found effective in Devlin et al. (2019); Conneau et al. (2020); Xue et al. (2021).

(OOD) data, whether in-domain or out-of-domain. We assess in-domain generalization by evaluating the *perplexity* of a model on the test set of each training data source, while OOD generalization is evaluated with unseen datasets. Furthermore, we evaluate DEPT's efficacy in building foundation models through downstream tasks: Natural Language Inference via MNLI (Williams et al., 2018), Question Answering via RACE (Lai et al., 2017), Sentence Similarity via STSB (Cer et al., 2017), and Sentence Classification via SST-2 (Socher et al., 2013) Since we use decoder-only models below the model-size threshold for in-context learning abilities (Brown et al., 2020), we follow Radford et al. (2018) for fine-tuning. The evaluation metrics are *accuracy* (MNLI, RACE, SST-2) and *Pearson correlation* (STSB). The full details are in Appendix E.

**Plasticity** refers to the model's ability to **quickly** and **effectively** adapt to a new domain, either to reach target performance with minimal steps or to achieve the highest possible performance. We evaluate the plasticity of DEPT models by training them on new data, such as a different domain or language, as well as the most heterogeneous subset of the training data, determined by the size of its local vocabulary within the shared global vocabulary (see Appendix A.2).

We assess training robustness and stability using the L2 norm of model parameters and activations. Model divergence in LLMs, as noted by the OPT (Zhang et al., 2022) and PaLM (Chowdhery et al., 2023) teams, correlates with rapid increases in activation norms, a trend also observed in vision transformers (Dehghani et al., 2023). While more common at large scales, this issue can arise in smaller transformers depending on learning rate suitability (Wortsman et al., 2024), which, like batch size, is influenced by the gradient noise scale for a given data distribution (McCandlish et al., 2018). Notably, all performance comparisons use optimized baseline hyperparameters (see Appendix A).

## 3.5 Continued Pre-training and Evaluation

Once pre-training is complete, some methods, including SPEC and ACT, lack a global embedding, while others, such as STANDARD pre-training, GLOB, and TRIM, include one. For ACT and SPEC (see Section 3.5), we enable a global (shared) embedding through multi-phase adaptive pre-training (Gururangan et al., 2020). This involves broad DEPT pre-training (Algorithm 1) followed by continued pre-training on another 15-19% of the **total** steps on a non-private dataset using a randomly initialized embedding matrix with a global vocabulary tailored to the specific corpus. For this phase, we use the tokenizer of Black et al. (2022) for English data and Xue et al. (2021) for multilingual data. These extra steps are applied to all models for fair comparison. While random initialization reveals the quality of the transformer body for all DEPT variants, we are also concerned with the independent effectiveness of GLOB and TRIM in building high-quality global embeddings compared to STANDARD methods. We perform the same 15-19% extra steps for this comparison, starting from pre-trained embeddings.

Unlike pre-training, this stage requires a sampling strategy. Since The Pile is curated for proportional sampling (Gao et al., 2021), we use it for multi-domain continued pre-training, while uniform sampling is applied to multilingual data to support low-resource languages.

## 4 Results

Our results show that DEPT improves transformer body generalization (Tables 3 and 4), enhancing robustness (Fig. 2), plasticity (Fig. 3), and downstream performance (Table 7) while bringing communication and memory costs reduction (Table 2).

## 4.1 DEPT Is Robust To Data Heterogeneity (RQ1)

Our experiments demonstrate DEPT's robustness to multilingual and multi-domain data heterogeneity. As shown in **Fig. 2**, DEPT resists activation divergence and model norm increases, which can halt perplexity improvements or cause divergence (Zhang et al., 2022; Chowdhery et al., 2023; Wortsman et al., 2024). When using the same local hyperparameters as the baselines, models trained with all DEPT variants maintain lower activation norms due to the regularization effects of OuterOpt (Algorithm 1). Learning rates for baselines are reduced for later comparisons to ensure convergence.

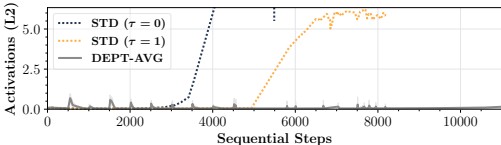 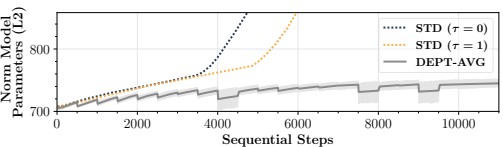

(a) `The Pile` pre-train, activation norms, 24-block   (b) `The Pile` pre-train, parameter norms, 24-block

Figure 2: Activations and model norms of STANDARD (`STD`) training versus `DEPT` (avg $\pm$ min/max) for a 350M model trained with identical local hyperparameters—prior to adjusting `STD` ($\tau = 0$) and `STD` ($\tau = 1$) (uniform and proportional sampling) to a lower learning rate. The `OuterOpt` of `DEPT` introduces regularization effects due to noise-injection (Lin et al., 2020), meta-learning (Nichol et al., 2018) characteristics, which constrain these sources (Zhang et al., 2022) of model divergence.

Table 2: Practical memory and communication costs for `DEPT`, where the total number of steps is $N = N_{\text{local}}T$ with $T$ the total number of iterations, and $\overline{\mathcal{V}_k}$ as the average vocabulary size across data sources. Standard pre-training requires a full in-memory embedding matrix for the global vocabulary while synchronizing gradients every step rather than every $N_{\text{local}}$ steps. All `DEPT` variants yield communication savings, with `GLOB` as the baseline. `TRIM` provides additional savings proportional to the gap between global and local vocabulary sizes, while `SPEC` further reduces costs by never communicating embeddings. For the full comparison, see Table 9.

| Type | #Blocks | Method | $N_{local}$ | $T$ | $\overline{|\mathcal{V}_k|} \pm \sigma$ | $\overline{|\mathcal{V}_k|} \times d_{\text{model}}$ | $\overline{\mathcal{M}_k}$ ($\downarrow$) | Per-step Comms Cost ($\downarrow$) |
|---|---|---|---|---|---|---|---|---|
| **Multilingual** | 12 | `STD` | $5 \times 10^3$ | 1 | 250 112 | 192M | 278M (1×) | 278M (1×) |
| **Multilingual** | 12 | `GLOB` | 500 | 10 | 250 112 | 192M | 278M (1×) | 0.56M (0.002×) |
| **Multilingual** | 12 | `TRIM` | 500 | 10 | 216 135 ± 27 160 | 166M | 252M (0.92×) | 0.5M (0.002×) |
| **Multilingual** | 12 | `SPEC` | 500 | 10 | 216 135 ± 27 160 | 166M | 252M (0.92×) | 0.17M (**0.0006×**) |
| **Multilingual** | 12 | `SPEC–OPT` | 500 | 10 | 50 257 ± 0 | 38.6M | 125M (**0.45×**) | 0.17M (**0.0006×**) |
| **Multilingual (1B)** | 24 | `STD` | $7 \times 10^3$ | 1 | 250 112 | 512.2M | 1.71B (1×) | 1.71B (1×) |
| **Multilingual (1B)** | 24 | `SPEC–OPT` | 500 | 14 | 50 257 ± 0 | 102.9M | 1.3B (**0.76×**) | 2.4M (**0.001×**) |

## 4.2 `DEPT` IMPROVES TRAINING EFFICIENCY (RQ2)

**Tables 1 and 2** show that `DEPT` significantly reduces average GPU memory and per-step communication costs compared to DDP. The 500× memory cost reduction from `GLOB` matches that of Local SGD, as it synchronizes gradients only every $N_{\text{local}}$ steps, allowing GPUs to operate independently in between. `TRIM` further improves memory and communication costs by reducing vocabulary size, shrinking the global embedding matrix by 8% to 32% for multilingual data and by 2% to 78% for `The Pile`, with the largest reduction (78%) achieved for the mathematics subset (see Appendix A.2 for precise vocab sizes). `SPEC` eliminates embedding-related communication, reducing costs by an additional 13% to 30% for multi-domain data and 34% for multilingual data. Finally, `DEPT` enables efficient training of billion-scale models (Fig. 4) on multilingual data, achieving a 714× reduction in communication costs (Table 2) and a 24% reduction in memory costs.

## 4.3 `DEPT` IMPROVES ZERO-SHOT GENERALIZATION (RQ3)

We show that `DEPT` variants significantly enhance transformer body generalization, outperforming STANDARD pre-training and active-forgetting (`ACT`) in: (a) perplexity on pre-training validation data, (b) perplexity on OOD validation data, and (c) downstream fine-tuning on `MNLI`, `RACE`, `STSB`. As detailed in Section 3.5, `DEPT` serves as the first stage of a multiphase adaptive pre-training pipeline, followed by continued pre-training on a non-private dataset. With pre-training data coalesced as in STANDARD training, Our results reflect performance after this phase is applied to baselines as well, ensuring embeddings process the same number of tokens. To gauge tokenizer effectiveness on a dataset, we report the unigram cross-entropy (`UNIGRAM–CE`) of the unigram model defined by the token frequencies, with higher values indicating a harder-to-model distribution (Tao et al., 2024)(see Appendix A.2.1). Overall, `DEPT` variants win $82.2\% = \frac{51}{62}$ of our main comparisons across `The Pile`, `MC4` and downstream tasks, producing generalizable and performant transformer bodies.

### 4.3.1 Transformer Body Generalization

Table 3: Validation perplexity (↓) for 24-block models trained on `The Pile` after **continued pre-training** with **proportional** sampling from **randomly-initialized** embeddings shows that `DEPT` improves performance across **all** data sources, outperforming baselines by 15.3% on average. `SPEC-OPT`, using an optimized vocabulary, outperforms `GLOB` on high `UNIGRAM-CE` sources.

| Name (UNIGRAM-CE) | DM (6.9) | EN (7.9) | EP (10) | FL (7.8) | GH (7.9) | CC (7.9) | PA (8.2) | SE (7.7) | PP (9.1) | WK (8.2) | AX (7.7) | UB (7.8) | PC (8) | NH (8.1) | GU (7.7) | HN (7.7) | UI-OOD (10) | AVG (8.1) |
|---|---|---|---|---|---|---|---|---|---|---|---|---|---|---|---|---|---|---|
| **STD** ($\tau = 0$) | 5.5 | 44.8 | 93.5 | 30.9 | 8.1 | 79.6 | 46.6 | 23.4 | 126.6 | 58.2 | 14.3 | 34.1 | 22.3 | 58.9 | 76.3 | 65.2 | 163.6 | 56 |
| **STD** ($\tau = 1$) | 5 | 30.6 | 49.5 | 20.6 | 6 | 56.2 | 30.9 | 16.8 | 81.2 | 39.1 | 11 | 23.7 | 16.1 | 39.3 | 54.6 | 46.9 | 99 | 36.9 |
| **ACT** | – | – | – | – | – | – | – | – | – | – | – | – | – | – | – | – | – | – |
| **GLOB** | 4.8 | 25.7 | 38.2 | **17.3** | 5.4 | **47.7** | **25.7** | **14.7** | 68.3 | **32.7** | 9.9 | 20 | 14 | **32.2** | 46.5 | 39.8 | 94.8 | 31.6 |
| **TRIM** | 4.8 | 27.3 | 39.5 | 18.5 | 5.6 | 51.2 | 27.8 | 15.4 | 71.8 | 35.1 | 10.3 | 21.7 | 14.8 | 35.1 | 49.1 | 42.2 | 95.7 | 33.3 |
| **SPEC** | 4.8 | 26.7 | 36.8 | 18.2 | 5.5 | 50.1 | 27.1 | 15.1 | 69.1 | 34.2 | 10.1 | 21.1 | 14.5 | 34.3 | 48.5 | 41.7 | 97.6 | 32.7 |
| **SPEC-OPT** | **4.7** | 25.9 | **35** | 17.5 | **5.4** | 48.3 | 26.1 | 14.7 | **66.6** | 32.8 | 9.9 | 20.4 | 14.1 | 32.9 | 47.3 | 40.5 | **88.6** | **31.2** |
| **Min Imp (%)** | 3.7 | 10.6 | 20.2 | 10.1 | 7.4 | 8.9 | 10.3 | 8.4 | 11.5 | 10.3 | 7 | 8.6 | 8.2 | 10.6 | 9.9 | 10 | 1.4 | 9.7 |
| **Max Imp (%)** | 4.2 | 15.7 | 29.3 | 16.3 | 11 | 15.1 | 16.9 | 12.9 | 17.9 | 16.5 | 10.6 | 15.7 | 13.3 | 18 | 14.7 | 15.2 | 10.5 | 15.3 |

Table 4: Validation perplexity (↓) for 12-block models trained on `MC4` using **continued pre-training** with **uniform sampling** from **randomly-initialized** embeddings. `DEPT` improves transformer performance across all languages, averaging a 17.3% gain for pre-train data and 20.8% on OOD sources. `SPEC` outperforms `GLOB` on high `UNIGRAM-CE` OOD data.

| | In-Distribution | | | | | | | | | Out-of-Distribution | | | |
|---|---|---|---|---|---|---|---|---|---|---|---|---|---|
| Name (UNIGRAM-CE) | ZH (9.8) | UR (10.5) | MS (9.2) | IT (7.7) | SR (10.5) | LA (9) | EN (7.5) | SW (10) | Avg (In-D) (9.3) | EL (14.4) | HI (13.9) | DE (9.7) | Avg (OOD) (12.6) |
| **STD** ($\tau = 0$) | 154.8 | 38.2 | 96.8 | 83.8 | 73.3 | 63 | 112.7 | 62.8 | 85.7 | 5660.8 | 4600.3 | 1339.2 | 3866.8 |
| **STD** ($\tau = 0.3$) | 129.5 | 34.5 | 88 | 75.4 | 65.2 | 56.3 | 103.7 | 56.8 | 76.2 | 4219.2 | 3996 | 1076.3 | 3097.1 |
| **STD** ($\tau = 1$) | 84.6 | 26.8 | 64.8 | 55.1 | 47.1 | 41.1 | 77.6 | 42.4 | 54.9 | 3340.3 | 2514.7 | 672.5 | 2175.8 |
| **ACT** | 96.1 | 28.8 | 71.3 | 60.4 | 52.3 | 44.9 | 85.6 | 46.3 | 60.7 | 2450.2 | 2412.5 | 715.9 | 1859.5 |
| **GLOB** | 67.7 | **22.4** | **53.7** | 46 | **38.6** | 33.9 | 65.4 | 35.2 | 45.4 | 2308.3 | 1676.5 | 559.5 | 1514.7 |
| **TRIM** | **67.7** | 22.8 | 55.2 | 47.5 | 39.7 | 35.1 | 67.2 | 36.3 | 46.4 | 2547.7 | 1911 | 567.4 | 1675.4 |
| **SPEC** | 69.5 | 23 | 55.4 | 47.8 | 40.3 | 34.7 | 68.1 | 36.3 | 46.9 | **2232.1** | **1578.8** | **544.7** | **1451.9** |
| **Min Imp (%)** | 17.8 | 14 | 14.5 | 13.4 | 14.6 | 14.6 | 12.2 | 14.3 | 14.4 | −4 | 20.8 | 15.6 | 10.8 |
| **Max Imp (%)** | 20 | 16.4 | 17.1 | 16.6 | 18.1 | 17.4 | 15.7 | 16.9 | 17.3 | 8.9 | 34.6 | 19 | 20.8 |

**Tables 3 and 4** present results where embedding matrices are initialized randomly. `DEPT` variants significantly outperform all baselines across validation sets for multilingual and multi-domain data sources, including high- and low-resource subsets. Min and max improvements, shown in the last two rows of the tables, compare the worst and best `DEPT` variants to the best-performing baseline. The best `DEPT` variant achieves an average performance improvement of 17.3% on `MC4` and 15.3% on `The Pile`, while even the worst variant shows improvements of 14.4% and 9.7%, respectively. `DEPT` wins $100\% = \frac{17}{17} = \frac{11}{11}$ comparisons for `The Pile` and `MC4`, respectively. For OOD data, `DEPT` variants outperform by 10-20% on average for `MC4` and 1.5-10.5% on `The Pile`, despite the high `UNIGRAM-CE` of OOD data, which makes it more difficult. This demonstrates that `DEPT` produces superior transformer bodies with better generalization. Notably, `TRIM` performs comparably to `GLOB` despite significant reductions in parameter counts and communication costs during pre-training, suggesting that out-of-vocabulary mistakes do not drastically impact performance. For downstream tasks, however, `TRIM` surpasses `GLOB` (Table 7). `SPEC` performs similarly to `GLOB` and `TRIM`, even without sharing token embeddings across data sources. The `SPEC-OPT` variant, trained with unique vocabularies and parameters for each `The Pile` data source, outperforms `GLOB` on datasets with high `UNIGRAM-CE` or those dissimilar to natural language, such as multilingual EP, math-heavy DM, code-based GH, and the high-`UNIGRAM-CE` dataset UI. For `MC4`, `SPEC` consistently outperforms on OOD datasets with high `UNIGRAM-CE`. These results hold across model sizes (see Table 12), and across sampling techniques (Table 10).

### 4.3.2 Pre-trained Embedding Generalization

**Tables 5 and 6** represent cases where the global embedding is initialized using the final global embedding obtained during pre-training, applicable only to the `GLOB` and `TRIM` variants. For `The Pile` (Table 5), both variants outperform their standard pre-training counterparts, achieving a 5.5% improvement in average accuracy and winning $\frac{12}{17}$ comparisons. Two of the lost comparisons, the small subsets EN and EP, are instead won when using uniform sampling (Table 11).

Table 5: Validation perplexity ($\downarrow$) for 24-block models trained on `The Pile` with **continued pre-training** using **proportional sampling** from **pre-trained embeddings**. `DEPT` wins $70\% = \frac{12}{17}$ comparisons with `GLOB` consistently outperforming `TRIM`. In Table 3, `DEPT` wins the remaining 5 due to its superior transformer body. Likewise, the `EN` and `EP` comparisons are won when using uniform sampling (Table 11) as embeddings become more refined on these smaller datasets.

| Name (UNIGRAM-CE) | DM (6.9) | EN (7.9) | EP (10) | FL (7.8) | GH (7.9) | CC (7.9) | PA (8.2) | SE (7.7) | PP (9.1) | WK (8.2) | AX (7.7) | UB (7.8) | PC (8) | NH (8.1) | GU (7.7) | HN (7.7) | UI-OOD (10) | AVG (8.1) |
|---|---|---|---|---|---|---|---|---|---|---|---|---|---|---|---|---|---|---|
| **STD** ($\tau=0$) | 4.4 | 13.8 | 15.6 | 14.9 | 5.1 | 41.8 | 20.7 | 13 | 38.3 | 26.8 | 9.5 | 17.1 | 12.7 | 23.4 | 37.2 | 30.9 | 54.1 | 22.3 |
| **STD** ($\tau=1$) | 4.5 | 19.9 | 21.9 | 13.3 | **4.5** | 37 | 19.7 | 11.6 | 47.8 | 24.5 | 8.5 | 16.2 | 11.5 | 25 | 36.4 | 31.7 | 54.3 | 22.8 |
| **GLOB** | 4.5 | 17 | 16.1 | **13.2** | 4.5 | **34.5** | **17.9** | **11.2** | **37.8** | **22.4** | **8.4** | **14.4** | **11** | **20.6** | **35.5** | **28.3** | 61.2 | **21.1** |
| **TRIM** | 4.6 | 20.5 | 23 | 13.9 | 4.6 | 38 | 20.2 | 12 | 49.9 | 25.1 | 8.7 | 16.6 | 11.8 | 25.7 | 38 | 32.9 | 56.8 | 23.7 |
| **Min Imp (%)** | −3 | −48.7 | −46.9 | −3.9 | −3.5 | −2.7 | −2.7 | −3.4 | −30.1 | −2.6 | −2.9 | −2.7 | −2.6 | −9.6 | −4.3 | −6.4 | −13.1 | −6 |
| **Max Imp (%)** | −1.2 | −23.6 | −3 | **0.9** | −0.8 | **6.8** | **9** | **3.4** | **1.4** | **8.4** | **0.9** | **11** | **4** | **12.3** | **2.6** | **8.4** | −5 | **5.5** |

Table 6: Validation perplexity ($\downarrow$) for 12-block models trained on `MC4` using **continued pre-training** with **uniform sampling** from pre-trained embeddings. `DEPT` achieves a 6.4% improvement in average perplexity for in-distribution data but slightly underperforms for OOD data, winning $50\% = \frac{4}{8}$ of in-distribution and $33\% = \frac{1}{3}$ of OOD comparisons. In Table 4, `DEPT` wins the remaining cases due to a better transformer body.

| Name (UNIGRAM-CE) | In-Distribution | | | | | | | | | Out-of-Distribution | | | |
|---|---|---|---|---|---|---|---|---|---|---|---|---|---|
| | ZH (9.8) | UR (10.5) | MS (9.2) | IT (7.7) | SR (10.5) | LA (9) | EN (7.5) | SW (10) | Avg (In-D) (9.3) | EL (14.4) | HI (13.9) | DE (9.7) | Avg (OOD) (12.6) |
| **STD** ($\tau=0$) | 57.8 | 21 | 46.5 | 40 | 33.6 | 29.4 | 57.5 | 30.3 | 39.5 | 1698.8 | 1365.7 | 385.5 | 1150 |
| **STD** ($\tau=0.3$) | 45.5 | 20.6 | 41.5 | 31 | **31.7** | **29.3** | 46.1 | 31.1 | 34.6 | **1419.4** | 1087.6 | 321.9 | **943** |
| **STD** ($\tau=1$) | 44.4 | 23.9 | 44.3 | **25.2** | 36.5 | 33.4 | **38.3** | 36.4 | 35.3 | 1583.6 | 1299.5 | **285.5** | 1056.2 |
| **GLOB** | **40.1** | **15.5** | **30.1** | 39.6 | 39 | 29.7 | 40.5 | **24.6** | **32.4** | 1737.3 | **823.4** | 335.1 | 965.3 |
| **TRIM** | 41.9 | 16.2 | 31.3 | 41.3 | 40.8 | 30.8 | 42 | 25.6 | 33.7 | 1725 | 855.2 | 345.6 | 975.3 |
| **Min Imp (%)** | 5.6 | 21.1 | 24.7 | −64 | −28.7 | −5.1 | −9.7 | 15.5 | **2.5** | −22.4 | 21.4 | −21.1 | −3.4 |
| **Max Imp (%)** | 9.7 | 24.4 | 27.6 | −57.4 | −22.8 | −1.2 | −5.8 | 18.7 | **6.4** | −21.5 | **24.3** | −17.4 | −2.4 |

Furthermore, `DEPT` consistently outperforms when starting from random embeddings due to its superior transformer body. Thus, we argue that differences in performance compared to results in Section 4.3.1 are primarily driven by variations in embedding sampling ratios. For `MC4` (Table 6), `DEPT` wins $\frac{4}{8}$ comparisons for in-distribution data and $\frac{1}{3}$ for OOD data, providing disproportionate benefits for the low-resource `UR` and `SW` languages. These languages have very high `UNIGRAM-CE` values, indicating that the global shared tokenizer, trained with temperature-weighted sampling, underserve them. Switching to proportional sampling during continued pre-training improves performance on high-resource lan-

Table 7: The performance on downstream tasks ($\uparrow$), following continued pre-training, shows that `DEPT` models achieve $3\% - 7.5\%$ relative improvements over the baselines, with `TRIM` delivering the best results. `DEPT` consistently outperforms baselines. For the full results see Table 21.

| Name | Random Init | | | |
|---|---|---|---|---|
| | RACE (ACC) | MNLI (ACC) | STSB (PC) | SST2 (ACC) |
| **STD** ($\tau=0$) | 0.50 | 0.60 | 0.66 | 0.79 |
| **STD** ($\tau=1$) | 0.46 | 0.68 | 0.73 | 0.81 |
| **ACT** | 0.45 | 0.66 | 0.73 | 0.80 |
| **GLOB** | 0.51 | **0.72** | 0.78 | 0.83 |
| **TRIM** | **0.53** | 0.71 | 0.78 | 0.83 |
| **SPEC** | 0.52 | 0.71 | **0.79** | 0.81 |
| **SPEC-OPT** | 0.51 | 0.69 | 0.77 | **0.85** |
| **Min Imp (%)** | 2.9% | 4.6% | 5.9% | −0.7% |
| **Max Imp (%)** | 5.8% | 6.1% | 7.5% | 4.1% |

guages, winning `EN`. Similarly to `The Pile`, the other comparisons are all won when starting from random embeddings. Thus, while `DEPT` may benefit the transformer body, care must be taken to design an appropriate continued pre-training pipeline to effectively fine-tune the embeddings.

### 4.3.3 DOWNSTREAM GENERALIZATION

**Table 7** presents the downstream performance of 24-block `DEPT` models pre-trained and continued pre-trained (with uniform sampling) on `The Pile`. `DEPT` models consistently outperform the baselines, regardless of initialization, with `TRIM` achieving the best results and `SPEC` matching `GLOB` in wins. Despite occasional losses to `GLOB` in language modeling, we speculate that the restricted vocabulary of `TRIM` forces it to adapt to language shifts, improving generalization, akin to `ACT`'s re-initialization but more effective. While `ACT` performs better on downstream tasks than on language modeling (Chen et al., 2023), it is outperformed by `DEPT`. `DEPT` leverages inherent aggregation noise to develop robust parameters without artificial re-initialization, ensuring that parameter updates are not discarded and avoiding the waste of compute cycles.

### 4.4 `DEPT` IMPROVES MODEL PLASTICITY (RQ4)

Finally, we investigate how plastic `DEPT` models are in adapting to either a new data source or to the most heterogeneous subset of the pre-training set. Figure 3 shows the perplexity adaptation plots when starting from a random initialization on the full pre-training set (serving as a baseline), the data source with the smallest vocabulary (`SW`), or new languages (`HI`, `DE`). `DEPT` variants are always the fastest to adapt to each data source and provide the lowest final perplexity; for the full pre-training set, we use perplexity taken over all language validation sets.

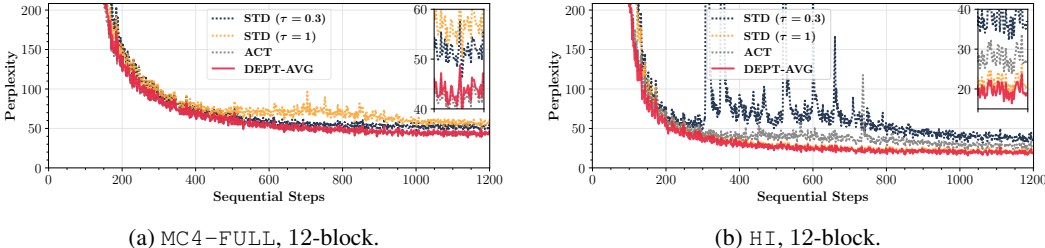

(a) `MC4-FULL`, 12-block.          (b) `HI`, 12-block.

Figure 3: Adaptation curves starting from a randomly initialized matrix. `DEPT` variants are always stable in their convergence, reaching the **lowest perplexity** for the full dataset and the out-of-distribution language (`HI`). It is also always the **fastest** to adapt, full results available in Figure 5

## 5 RELATED WORK

Large language models (LLMs) exhibit cross-lingual alignment due to "incidental bilingualism" (Briakou et al., 2023) and cross-lingual data sharing (Choenni et al., 2023). Expanding multilingual data during pre-training can enhance language diversity (Scao et al., 2022) but often results in uneven performance due to data imbalance and low-resource degradation (Ding et al., 2024; Lai et al., 2023). Supervised parallel data (e.g., XLM (Conneau & Lample, 2019), PaLM2 (Anil et al., 2023)), Knowledge Transfer (Zhang et al., 2023; Wang et al., 2023), and Domain Adaptation (Huang et al., 2024) face challenges in low-resource settings (Chang et al., 2023b; Li et al., 2024), with risks like training instability and catastrophic forgetting (Kirkpatrick et al., 2017). This motivates our novel pipeline, focusing on language heterogeneity, generalization, and plasticity. Vocabulary construction is crucial in multilingual pre-training. Techniques include tokenization with a temperature setting (Devlin et al., 2019) and language-clustered vocabularies (Chung et al., 2020), though the latter requires predefined clusters. Active forgetting (Chen et al., 2023), a related approach, enhances model plasticity by periodically re-initializing embeddings, easing adaptation to new languages.

## 6 CONCLUSION

We investigated pre-training Language Models (LMs) under data heterogeneity, proposing an efficient and robust pipeline, `DEPT`, which supports training under diverse data sources while mitigating *Negative Interference* and the *Curse of Multilinguality*. The core of `DEPT` is decoupling the embedding space from the transformer body during pre-training, offered in three variants with varying degrees of separation. Experiments showed that `DEPT` (1) allows training across heterogeneous data efficiently, (2) reduces the memory footpring of token embedding matrices by $4-5\times$, (3) improves model generalization and plasticity with lower perplexity on validation and out-of-distribution test datasets, and (4) supports custom vocabularies per data source, enabling vocabulary agnostic federated pre-training, which we have tested up to billion-scale models and intend to push further.

### 6.1 LIMITATIONS & FUTURE WORK

`DEPT` offers a *pre-training* framework intended to precede further adaptation or fine-tuning. However, `DEPT` models require a final global embedding for practical use. The `GLOB` and `TRIM` variants provide this at the end of pre-training, while `SPEC` does not, suggesting future work on embedding generation methods, such as zero-shot embedding transfer (Mosin et al., 2023), vocabulary matching (Xu et al., 2024) and model stitching (Moschella et al., 2023).

ACKNOWLEDGMENTS

All costs for the computation used for this work was funded by Flower Labs, and the research conducted by a team of researchers from Flower Labs and The University of Cambridge. Support for university-based researchers came from a variety of sources, but in particular, the following funding organizations are acknowledged: the European Research Council (REDIAL), the Royal Academy of Engineering (DANTE), and the Ministry of Education of Romania through the Credit and Scholarship Agency.

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

# A    EXPERIMENTAL DETAILS

## A.1    MODEL ARCHITECTURES AND HYPERPARAMETERS

Table 8 presents the vocabulary-agnostic hyperparameters of our decoder-only models, while Table 9 details vocabulary sizes, `DEPT`-specific parameters, memory costs, and communication costs. Standard pre-training pipeline parameters were chosen based on the recommendations of Hoffmann et al. (2022) and MosaicML, except for the billion-scale model, where we aligned with the recent state-of-the-art (SOTA) for English federated pre-training by Sani et al. (2024). We always use a gradient clipping norm of 1 and `ALiBi` (Press et al., 2022) positional embeddings.

During continued pre-training, for models initialized randomly, we begin with $\eta_{\max}$ and decay over $N_{\mathrm{CT}}$ learning steps, allowing quick embedding matrix learning without requiring another full training pass, as is common in language rewiring (Artetxe et al., 2020). When using pre-initialized models, we start from $\eta_{\max}/2$ since both the model and embeddings are reasonably well-trained.

Importantly, the only parameter changed between `DEPT` models and baselines is the learning rate $\eta_{\max}$. We use the same learning rate to contrast convergence properties for comparisons in Fig. 2. We tune the baselines' learning rate for later comparisons to ensure they perform the same number of training steps, selecting the best checkpoint for a baseline across all experiments. Except for tuning the learning-rate, `DEPT` models always use the same hyperparameters as the baselines during local training.

Table 8: Architectural details and vocabulary-independent hyperparameters of our models. The number of transformer blocks is denoted by #Blocks, the number of attention heads by #Heads, and the expansion ratio refers to the ratio of the hidden dimension in the feedforward layers. The total number of model parameters is $\mathcal{M}$, the vocabulary size is $|\mathcal{V}|$, and the model embedding dimension is $d_{\mathrm{model}}$. We train standard decoder-only transformers whose body ranges in size from 86.4M to 1.2B independent of embeddings. As we see in Table 9, the size of the embedding matrix can change the model size drastically. Our batch size is $|\mathcal{B}|$ while $|S_t|/|S|$ is our sampling ratios for the various data sources. The $\beta_1, \beta_2$ pair are AdamW parameters while the $S_c$ tuple represents the parameters of the cosine scheduler that we use, including the decay alpha $\alpha$, the decay period $\eta_{\max}$, and the total number of sequential steps $N$. Finally, we show the number of continued pre-training steps $N_{\mathrm{CT}}$ that we use, representing 15% of total steps for the 298M model and 19.3% for the 86.4M model. All of our models use a sequence length of 2048. We followed the hyperparameters of Sani et al. (2024) for the billion-scale federated pre-training. We report the tuned $\eta_{\max}$, for each baseline according to Appendix A.1.2, $\eta_{\max}^{\mathrm{STD}(\tau=0)}, \eta_{\max}^{\mathrm{STD}(\tau=0.3)}, \eta_{\max}^{\mathrm{STD}(\tau=1)}$, we find that the embedding resting allows `ACT` to use the same $\eta_{\max}$ as `DEPT`.

| Type | #Blocks | $d_{\mathrm{model}}$ | $\mathcal{M} - |\mathcal{V}| \times d_{\mathrm{model}}$ | #Heads | Exp. Ratio | $|\mathcal{B}|$ | $|S_t|/|S|$ | $(\beta_1, \beta_2)$ | $S_C(\alpha, \eta_{max}, N)$ | $N_{CT}$ | $\eta_{\max}^{\mathrm{STD}(\tau=0)}$ | $\eta_{\max}^{\mathrm{STD}(\tau=0.3)}$ | $\eta_{\max}^{\mathrm{STD}(\tau=1)}$ |
|---|---|---|---|---|---|---|---|---|---|---|---|---|---|
| Multi-domain | 12 | 768 | 86.4M | 12 | 4 | 256 | 4/16 | (0.9, 0.95) | $(10^{-1}, 6.0 \times 10^{-4}, 5 \times 10^3)$ | $1.2 \times 10^3$ | $4.5 \times 10^{-4}$ | $4.5 \times 10^{-4}$ | $5.0 \times 10^{-4}$ |
| Multi-domain | 24 | 1024 | 298.5M | 16 | 4 | 256 | 4/16 | (0.9, 0.95) | $(10^{-1}, 3 \times 10^{-4}, 13.5 \times 10^3)$ | $2.4 \times 10^3$ | $1.5 \times 10^{-4}$ | $2 \times 10^{-4}$ | $2 \times 10^{-4}$ |
| Multilingual | 12 | 768 | 86.4M | 12 | 4 | 256 | 3/8 | (0.9, 0.95) | $(10^{-1}, 6 \times 10^{-4}, 5 \times 10^3)$ | $1.2 \times 10^3$ | $4 \times 10^{-4}$ | $4 \times 10^{-4}$ | $4.5 \times 10^{-4}$ |
| Multilingual | 24 | 2048 | 1.2B | 16 | 4 | 512 | 3/8 | (0.9, 0.95) | $(10^{-1}, 2 \times 10^{-4}, 7 \times 10^4)$ | - | $1 \times 10^{-4}$ | $1 \times 10^{-4}$ | $1.5 \times 10^{-4}$ |

We had to select a particular sampling ratio for the continued pre-training using the full pre-training set rather than a single language or domain. Due to its high heterogeneity, we default to uniform sampling for `MC4` in these cases. In contrast, for `The Pile`, we preferred proportional sampling as the dataset is entirely in English and has already had its data sources upsampled/downsampled based on usefulness. We also provide results using the alternative sampling policy in Appendix B.

### A.1.1    SOFTWARE AND HARDWARE

Our software is based on the MosaicML composer (Databricks, 2024) library for LLM pre-training and the open-source Flower (Beutel et al., 2022) framework for federated learning. Crucially, we heavily rely on the MosaicML hyperparameters and infrastructure for our `InnerOPT`, making no changes to it after our embedding-matrix manipulation from Algorithm 1 has been performed. For the standard baselines, we ran them on a completely unmodified version of the MosaicML codebase (beyond using our data), which has been independently verified by thousands of users and used to submit accepted conference publications (Blakeney et al., 2024).

Table 9: Practical memory and communication costs for `DEPT`, where the total number of steps is $N = N_{\text{local}}T$ with $T$ the total number of iterations, and $\overline{\mathcal{V}_k}$ as the average vocabulary size across data sources. Standard pre-training requires a full in-memory embedding matrix for the global vocabulary while synchronizing gradients every step rather than every $N_{\text{local}}$ steps. All `DEPT` variants yield communication savings, with `GLOB` as the baseline. `TRIM` provides additional savings proportional to the gap between global and local vocabulary sizes, while `SPEC` further reduces costs with or without optimized vocabularies by never communicating the token or positional matrices.

| Type | #Blocks | Method | $N_{local}$ | $T$ | $\overline{\|\mathcal{V}_k\|} \pm \sigma$ | $\overline{\|\mathcal{V}_k\|} \times d_{\text{model}}$ | $\overline{\mathcal{M}_k}$ ($\downarrow$) | Per-step Comms Cost ($\downarrow$) |
|---|---|---|---|---|---|---|---|---|
| **Multilingual** | 12 | `STD` | $5 \times 10^3$ | 1 | 250 112 | 192M | 278M (1×) | 278M (1×) |
| **Multilingual** | 12 | `GLOB` | 500 | 10 | 250 112 | 192M | 278M (1×) | 0.56M (0.002×) |
| **Multilingual** | 12 | `TRIM` | 500 | 10 | 216 135 ± 27 160 | 166M | 252M (0.92×) | 0.5M (0.002×) |
| **Multilingual** | 12 | `SPEC` | 500 | 10 | 216 135 ± 27 160 | 166M | 252M (0.92×) | 0.17M (**0.0006×**) |
| **Multilingual** | 12 | `SPEC-OPT` | 500 | 10 | 50 257 ± 0 | 38.6M | 125M (**0.45×**) | 0.17M (**0.0006×**) |
| **Multilingual-B** | 24 | `STD` | $7 \times 10^3$ | 1 | 250 112 | 512.2M | 1.71B (1×) | 1.71B (1×) |
| **Multilingual-B** | 24 | `SPEC-OPT` | 500 | 14 | 50 257 ± 0 | 102.9M | 1.3B (**0.76×**) | 2.4M (**0.001×**) |
| **Multi-domain** | 12 | `STD` | $5 \times 10^3$ | 1 | 50 257 | 38.6M | 125M (1×) | 125M (1×) |
| **Multi-domain** | 12 | `GLOB` | 500 | 10 | 50 257 | 38.6M | 125M (1×) | 0.25M (0.002×) |
| **Multi-domain** | 12 | `TRIM` | 500 | 10 | 45 554 ± 9462 | 35M | 121M (**0.97×**) | 0.24M (0.002×) |
| **Multi-domain** | 12 | `SPEC` | 500 | 10 | 45 554 ± 9462 | 35M | 121M (**0.97×**) | 0.17M (**0.001×**) |
| **Multi-domain** | 24 | `STD` | $13.5 \times 10^3$ | 1 | 50 257 | 51.4M | 350M (1×) | 350M (1×) |
| **Multi-domain** | 24 | `GLOB` | 500 | 27 | 50 257 | 51.4M | 350M (1×) | 0.7M (0.002×) |
| **Multi-domain** | 24 | `TRIM` | 500 | 27 | 45 554 ± 9462 | 46.6M | 345.2M (**0.97×**) | 0.69M (**0.002×**) |
| **Multi-domain** | 24 | `SPEC` | 500 | 27 | 45 554 ± 9462 | 46.6M | 345.2M (**0.97×**) | 0.6M (0.002×) |

In terms of hardware, the low communication properties of `DEPT` allowed us to run experiments via a mixture of loaned resources from separate cloud providers. Over the course of our experimentation, we used various machines equipped with either 1 H100 or 1 A100 GPU in the USA, Canada, and Europe, which turned out to be more cost-effective. We rented machines with 4-8 H100 GPUs for the centralized baselines since we could not use Distributed Data Parallelism techniques over low-bandwidth internet connections. When the standard training baseline has a sufficiently low learning rate to converge, the difference in training time is driven by three factors.

First, the throughput achieved by individual workers: for `GLOB`, this should be identical to standard pre-training as the model in memory remains unchanged. For `TRIM` and `SPEC`, the reduced memory requirements may allow increasing the device micro-batch size in certain scenarios (but not the global batch size, which heavily influences optimization properties). This depends heavily on the hardware; for example, in DeepMind Mathematics workloads, `TRIM` or `SPEC` can double the device micro-batch size, and similarly for `SPEC-OPT` in the case of multilingual data.

Second, the communication topology significantly impacts wall clock time. For instance, in a 10 Gbps bandwidth connection using Ring AllReduce for aggregation across workers, `DEPT` can reduce training time by 33% for a 1 billion parameter model. In cases with a very fast connection, such as InfiniBand, the training time difference is primarily determined by throughput differences.

Third, the number of local data sources and the number of available workers impact the total training time, for `DEPT` we always scale the number of workers to match the number of data sources exactly.

### A.1.2 Hyperparameter Tuning Methodology

Given that MosaicML provides hyperparameter-tuned models on the C4 (Raffel et al., 2020) dataset, we use their learning rate schedule and number of training steps as a starting point. In the case of `DEPT`, we find that we can always use the MosaicML parameters since the `OuterOpt` application of `DEPT` acts as a regulariser via noise-injection (Lin et al., 2020) and meta-learning effects (Nichol et al., 2018). This makes `DEPT` models highly unlikely to diverge, even under extreme data heterogeneity and without a shared input or output space. In the case of standard training baselines, we gradually lower the learning rate, starting from the one reported in Table 8.

We begin with the maximum learning rate $\eta_{\max}$ and systematically reduce it on a coarse grid in intervals of $0.5 \times 10^{-5}$:

$$\eta = \eta_{\max} - 0.5k \times 10^{-5}, \quad k \in \{0, 1, 2, \dots, K\},$$

where $k$ represents the step index, and $K$ is chosen such that $\eta > 0$ at the final step. Given that the length of the cosine cycle is directly extrapolated from known scaling laws on the number of tokens that the model needs to train on for compute-optimality (Hoffmann et al., 2022), approximately 20 tokens per parameter, we stop as early as we find a learning rate that can complete the entire cosine schedule. Then, we choose the best-performing checkpoint, according to validation perplexity, across all experiments. We report these values in Table 8.

This hyperparameter search does not cover all possible relevant parameters; given enough resources, we would also tune the gradient clipping norm. Furthermore, we could tune the batch size using the empirical model of large-batch training proposed by McCandlish et al. (2018). Given that the appropriate learning rate depends on the chosen batch size and the desired target loss, such an optimization would require hundreds of experiments across all baselines to find an optimal configuration.

### A.1.3 Adapting Active Forgetting

To implement the active forgetting baseline (Chen et al., 2023), `ACT`, we had to adapt the methodology to decoder-only models, which train with far fewer steps. To achieve this, we use a forgetting frequency of 500 steps, equal to `DEPT`'s $N_{\text{local}}$. We also use a cosine scheduler for the body with the same parameters as shown in Table 8; however, we schedule the embedding matrix independently across the 500 steps using the same scheduler but setting $\eta'_{\text{max}} = 500$. Finally, we selected the checkpoint with the lowest validation perplexity for continued pre-training in a forgetting cycle.

### A.2 Data Sources

We quantify the lexical heterogeneity of a dataset based on *lexical similarity* between data sources. A simple similarity measure is the size of the intersection of subwords between vocabularies. The smaller the intersection, the more dissimilar the vocabularies, and thus, the more challenging it becomes to train a shared tokenizer effectively across different domains or languages. For this section, we use the size of local vocabulary as a subset of the global vocabulary as a proxy, with smaller local vocabulary indicating that global tokenization does not serve a particular data source well.

Our default global tokenizer for multilingual data is that proposed by Xue et al. (2021), with $\mathcal{V} = 250\,112.0$ tokens. Owing to its diverse pre-training, the mT5 (Xue et al., 2021) tokenizer is a robust default choice, employed in recent works such as project Aya (Üstün et al., 2024). However, its coverage of hundreds of languages does come with many shortcomings relating to the capacity allocated to each language. To showcase these challenges, we carefully selected languages from distinct families in the `MC4` subset, including `English (EN)`, `Italian (IT)`, `Serbian (SR)`, `Swahili (SW)`, `Urdu (UR)`, `Latin (LA)`, `Chinese (ZH)`, and `Malay (MS)`. The corresponding vocabulary sizes of our languages are as follows: $\{247\,720, 211\,332, 208\,391, 170\,984, 188\,002, 220\,757, 240\,566\}$. Among these, `Swahili (SW)` is the most heterogeneous, as determined by its small subset of $170\,984$ tokens.

Our global tokenizer for English data was trained on `The Pile` (Gao et al., 2021) and proposed by Black et al. (2022) with $\mathcal{V} = 50\,257$ tokens. We selected `The Pile` as our multidomain dataset for several reasons. `The Pile` is a diverse, large-scale dataset specifically designed for training large language models (LLMs). Its diversity spans domains such as scientific papers, news, books, and web content, providing a comprehensive foundation for capturing varied linguistic patterns. Among the various subsets of `The Pile`, *DM Mathematics* stands out as the most heterogeneous. This subset contains only $11,090$ tokens from the global vocabulary, significantly fewer than other subsets. Here are the sizes of other subsets in terms of their unique tokens from the global vocabulary: $\{49\,362, 49\,783, 46\,766, 49\,469, 49\,700, 47\,865, 48\,720\}$ $\{11\,090, 44\,249, 42\,957, 44\,432, 49\,992, 49\,841, 47\,687, 49\,961, 46\,825\}$. While this indicates much lower heterogeneity than in multilingual settings, vocabulary choice may still impact highly specialized model capabilities such as mathematical reasoning.

### A.2.1 Tokenization Considerations

One of the major challenges when representing multiple data sources with a single tokenizer is *vocabulary dilution*. To maximize coverage, a tokenizer that aims to cover multiple languages or

domains often needs to adopt many short subwords. This increases the *tokenizer fertility* (i.e., the number of tokens produced per unit of text) (Rust et al., 2021) and also raises the overall *description length* — the total number of tokens required to represent the same data. This trade-off negatively affects the *compression ratio*, as the same amount of information requires more tokens, reducing the model's sample efficiency (Tao et al., 2024). When non-uniform sampling ratios are used during pre-training, high-resource languages tend to have better *fertility* than low-resource languages. This means high-resource languages are better represented in the vocabulary, and their tokens are more likely to be shared across the model's parameters, improving their performance. In contrast, low-resource languages suffer from poor fertility, where their unique vocabulary tokens are underrepresented, leading to worse performance. For example, `Swahili (SW)` and `Urdu (UR)` are low-resource languages that face these challenges. Our `SPEC` method allows us to avoid many of these issues by providing an optimized vocabulary to a data source at the cost of losing a shared vocabulary and updating several embedding matrices. An alternative approach is to cluster vocabularies (Chung et al., 2020) to obtain subword sharing between more relevant languages. However, this requires that participating data sources are known in advance, do not change significantly, and that the appropriate number of clusters is also known.

To account for the effectiveness of a tokenizer on a given language, we report unigram cross-entropy in our experiments, which represents how effective a simple unigram model based on the tokenizer is on that data source as a proxy for the effectiveness of the tokenization. If the unigram cross-entropy is high on a given data source, it is likely underserved by the tokenization. Thus, all improvements brought about by using a more complex language model must consider this baseline. It can also be used to compute unigram-normalized cross-entropy or perplexity, a language modeling performance metric that is comparable across different vocabulary sizes (Tao et al., 2024).

# B ADDITIONAL RESULTS

## B.1 LARGER MODEL

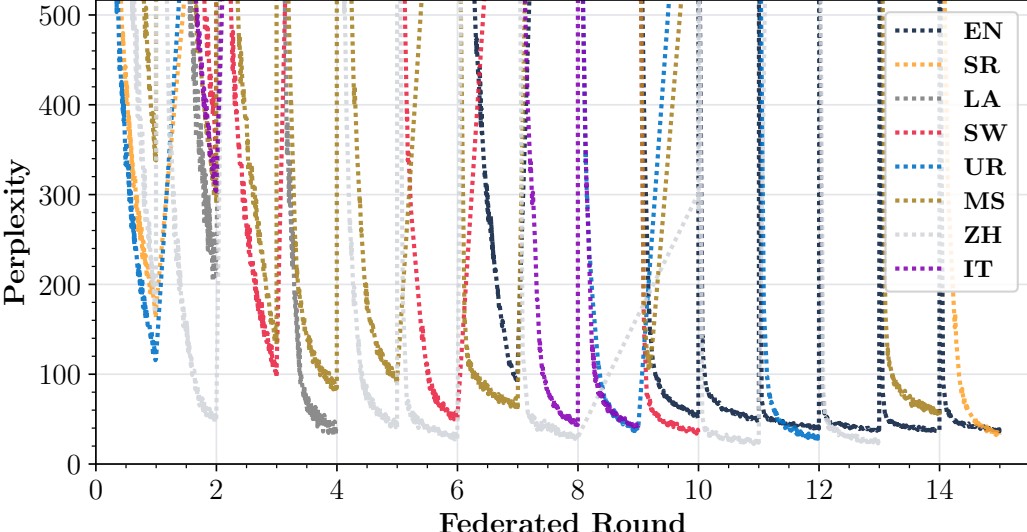

Figure 4: Convergence plot of our 1.3 billion model trained in a vocabulary agnostic federated fashion. For the initial rounds, we sample 4 data sources out of 8; after seeing most of the clients, we reduce the number to 2. We make sure only to introduce `EN` later into the experiment.

Figure 4 provides further insights into the performance of `DEPT` on a larger-scale experiment with a 1.3 billion-parameter model. In this setting, the model is trained in a vocabulary-agnostic, federated fashion with dynamic client subsampling. During the initial rounds, 4 out of 8 data sources are sampled, which is reduced to 2 after most clients have been processed. Importantly, `EN` is in-

troduced later in the training process to evaluate the model's cross-lingual transfer capabilities to this high-resource language. The plot illustrates that the transformer body, enabled by DEPT, effectively transfers knowledge across languages and domains, allowing newly introduced or previously stale data sources to converge to perplexity levels similar to their peers within one or two sampling rounds. This experiment underscores the feasibility and scalability of using DEPT for collaborative large-scale language model pre-training, even under extreme client subsampling and without prior knowledge of the underlying data distribution.

## B.2 PLASTICITY

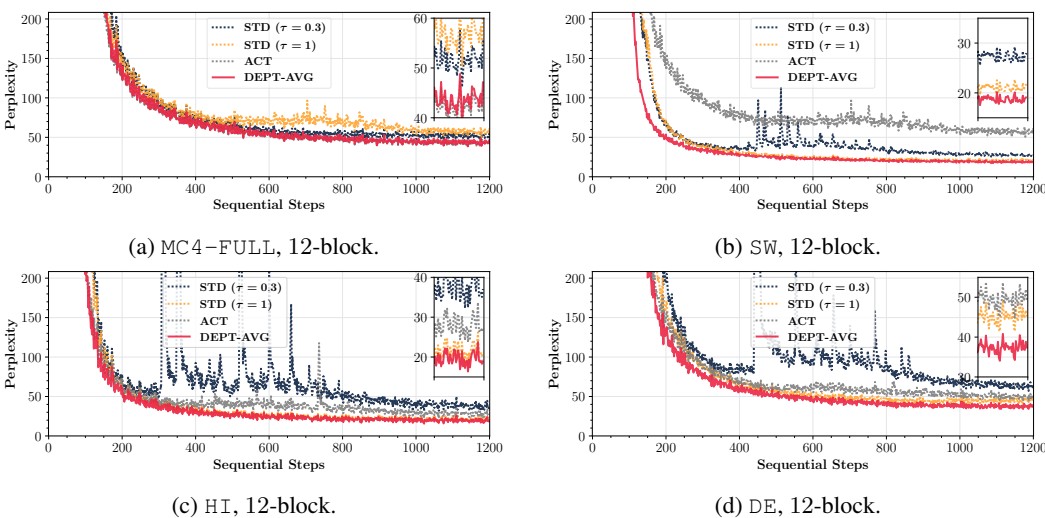

(a) MC4-FULL, 12-block.  (b) SW, 12-block.

(c) HI, 12-block.  (d) DE, 12-block.

Figure 5: Adaptation curves starting from a randomly initialized matrix. DEPT is always stable in its convergence, reaching the **lowest perplexity** for the pre-training distribution (MC4-FULL), for the lowest-resource languages in the distribution (SW), and for the two out-of-distribution languages (HI, DE). It is also always the **fastest** to adapt.

The results presented in Figure 5 demonstrate the robustness and adaptability of DEPT across various settings, completing the plot shown in Fig. 3. Specifically, DEPT consistently achieves the lowest perplexity across all scenarios: (1) the full pre-training distribution (MC4-FULL), (2) the lowest-resource language within the dataset (SW), and (3) two out-of-distribution languages (HI and DE). Furthermore, DEPT is not only effective in reaching convergence but also does so at a faster rate compared to other approaches. These results showcase its utility in a wide range of multilingual and domain-adaptive pre-training tasks; for example, if a new client were to be introduced in a federated setting, they show that the DEPT trained model could quickly adapt to its data distribution. Alternatively, multi-phase adaptive pre-training represents a distinct advantage in terms of data efficiency.

## B.3 IID DATA PERFORMANCE

In the case of IID data (represented by a random sharding of the C4 dataset), Fig. 6 shows that DEPT performs similarly to standard pre-training with the benefit of lower activation norms, indicating the potential for longer and more training.

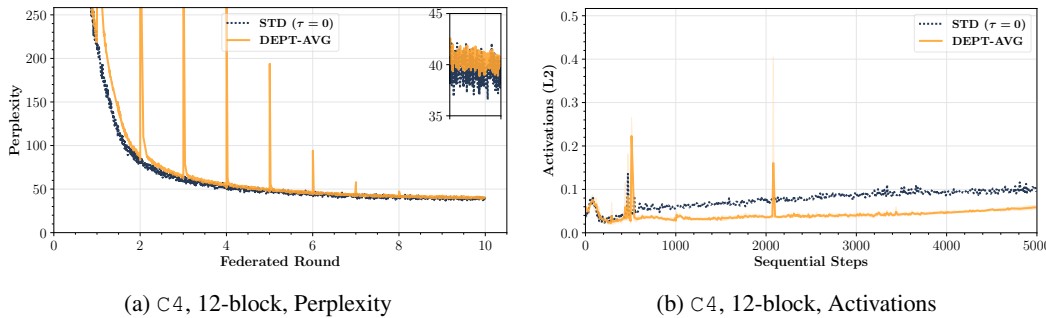

(a) `C4`, 12-block, Perplexity — (b) `C4`, 12-block, Activations

Figure 6: Perplexity (a) and activations (b) curves for `DEPT` versus uniform sampling on the IID `C4` dataset. `DEPT` models, outside temporary spikes caused by `OuterOpt`, perform similarly to standard pre-training regarding training perplexity. However, as seen from the activations, it still provides greater training stability with the potential of extending pre-training.

## B.4 ONE-SHOT GENERALIZATION

Table 10: Validation perplexity (↓) for our 24-block models trained on `The Pile` when using continued pre-training with uniform sampling starting from randomly-initialized embeddings. `DEPT` provides a better transformer body for **all** datasets, outperforming baselines by 17.5% on average.

| Name (UNIGRAM-CE) | DM (6.9) | EN (7.9) | EP (10) | FL (7.8) | GH (7.9) | CC (7.9) | PA (8.2) | EN (7.7) | PP (9.1) | WK (8.2) | AX (7.7) | UB (7.8) | PC (8) | NH (8.1) | GU (7.7) | HN (7.7) | UI-OOD (10) | AVG (8.1) |
|---|---|---|---|---|---|---|---|---|---|---|---|---|---|---|---|---|---|---|
| **STD** ($\tau = 0$) | 4.7 | 17.5 | 19.7 | 22.3 | 7 | 64.2 | 30.7 | 18.3 | 48.5 | 41.6 | 13 | 23.9 | 19 | 32.1 | 50.5 | 42 | 72.9 | 31.1 |
| **STD** ($\tau = 1$) | 5.1 | 24.1 | 27.1 | 31.4 | 9.2 | 86.6 | 43.4 | 24.6 | 64.4 | 58.4 | 16.5 | 32.7 | 25.1 | 44.7 | 66.7 | 55.8 | 141 | 44.5 |
| **ACT** | – | – | – | – | – | – | – | – | – | – | – | – | – | – | – | – | – | – |
| **GLOB** | 4.5 | 14.2 | 16.3 | 18 | 6.1 | 53.6 | 24.9 | 15.5 | 40.2 | 34 | 11.2 | 19.7 | 16 | 26.2 | 41.9 | 34.6 | 58.8 | 25.6 |
| **TRIM** | 4.5 | 14.8 | 16.7 | 19.1 | 6.4 | 56.6 | 26.4 | 16.3 | 42.1 | 36 | 11.7 | 21 | 16.9 | 27.7 | 43.7 | 36.2 | 66.1 | 27.2 |
| **SPEC** | 4.5 | 14.5 | 16.2 | 18.8 | 6.2 | 55.5 | 25.8 | 16 | 41.1 | 35.1 | 11.5 | 20.5 | 16.5 | 27.2 | 43.1 | 35.7 | 63.5 | 26.6 |
| **SPEC−OPT** | 4.6 | 15.2 | 16.9 | 19.4 | 6.4 | 57.1 | 26.5 | 16.4 | 42.5 | 35.9 | 11.9 | 21 | 16.5 | 27.8 | 44.9 | 37.1 | 60.4 | 27.1 |
| **Min Imp (%)** | 2.5 | 13.5 | 14.3 | 12.9 | 8.4 | 11.2 | 13.9 | 10.4 | 12.5 | 13.7 | 8.2 | 12.3 | 11.5 | 13.4 | 11 | 11.8 | 9.2 | 12.5 |
| **Max Imp (%)** | 4.8 | 19.1 | 18 | 19.3 | 13 | 16.5 | 19 | 15.3 | 17.1 | 18.3 | 13.5 | 17.6 | 16 | 18.6 | 17.1 | 17.6 | 19.4 | 17.5 |

## B.5 TRANSFORMER BODY GENERALIZATION

Table 10 shows the performance of `DEPT` on the `The Pile` dataset with a 24-block model trained from randomly initialized embeddings. Here, `DEPT` outperforms all baselines across all subsets, with average improvements of 17.5%.

## B.6 PRE-TRAINED EMBEDDING MATRIX GENERALIZATION

Table 11: Validation perplexity (↓) for our 24-block models trained on `The Pile` when performing continued pre-training with uniform sampling starting from a pre-trained embedding matrix. `DEPT` wins 10 out of 17 comparisons with `TRIM` always outperforming `GLOB`. When comparing against Tables 5 and 10, we can observe that `DEPT` wins the complementary comparisons when starting from random embeddings or when using proportional sampling with the pre-trained embedding matrices. This indicates that baselines always have a **worse** transformer body, with sampling ratios heavily impacting the effectiveness of embeddings for a given dataset.

| Name (UNIGRAM-CE) | DM (6.9) | EE (7.9) | EP (10) | FL (7.8) | GH (7.9) | CC (7.9) | PA (8.2) | SE (7.7) | PP (9.1) | WK (8.2) | AX (7.7) | UB (7.8) | PC (8) | NH (8.1) | GU (7.7) | HN (7.7) | UI-OOD (10) | AVG (8.1) |
|---|---|---|---|---|---|---|---|---|---|---|---|---|---|---|---|---|---|---|
| **STD** ($\tau = 0$) | 4.3 | 11.1 | 13.4 | 15 | 5.5 | 44.4 | 20.4 | 13.2 | 34 | 27.1 | 10.1 | 16.9 | 13.6 | 21.6 | 35.2 | 29.1 | 51.6 | 21.6 |
| **STD** ($\tau = 1$) | 4.3 | 12.6 | 15.8 | 13.5 | 4.7 | 38.8 | 19.2 | 11.7 | 36.8 | 24.7 | 8.8 | 16 | 12 | 21.7 | 34.2 | 28.6 | 43.1 | 20.4 |
| **GLOB** | 4.5 | 12.1 | 13.8 | 15.6 | 5.5 | 41.4 | 19.4 | 12.9 | 33.5 | 25.6 | 10.1 | 15.6 | 13.3 | 19.8 | 38.1 | 29.2 | 58.2 | 21.7 |
| **TRIM** | 4.4 | 10 | 11.3 | 14.8 | 4.9 | 37.3 | 18.2 | 11.8 | 30.2 | 23 | 9.8 | 15.3 | 12.8 | 19 | 32.9 | 26.7 | 47.6 | 19.4 |
| **Min Imp (%)** | −4.5 | −8.6 | −2.9 | −15.6 | −17.8 | −6.8 | −0.8 | −9.6 | 1.2 | −3.5 | −14 | 2 | −10.2 | 8.1 | −11.5 | −2.2 | −35.1 | −6.4 |
| **Max Imp (%)** | −1.2 | 10.1 | 15.4 | −9.5 | −5.1 | 3.8 | 5.1 | −0.3 | 11.1 | 7.1 | −10.9 | 4.2 | −6 | 12.1 | 3.9 | 6.6 | −10.4 | 4.8 |

When continuing pre-training with pre-trained embedding matrices, as shown in Table 11, `DEPT` secures 10 out of 17 wins, with `TRIM` consistently outperforming `GLOB`. A comparison with Tables 10 and 5 reveals that `DEPT` also outperforms in other scenarios, whether starting from random

Table 12: Validation perplexity (↓) for our 12-block models trained on `The Pile` when performing continued pre-training starting from a randomly-initialized embedding matrix. DEPT can train a superior transformer body, outperforming all baselines across all subsets by up to 28%.

| Name (UNIGRAM-CE) | NH (8.1) | GH (7.9) | PA (8.2) | UB (7.8) | FL (7.8) | EE (7.9) | EP (10) | WK (8.2) | CC (7.9) | SE (7.7) | PC (8) | PP (9.1) | DM (6.9) | AX (7.7) | GU (7.7) | HN (7.7) | UI-OOD (10) | AVG (8.1) |
|---|---|---|---|---|---|---|---|---|---|---|---|---|---|---|---|---|---|---|
| STD ($\tau = 0$) | 63.0 | 12.1 | 61.9 | 44.1 | 45.2 | 36.3 | 47.9 | 81.8 | 115.9 | 33.2 | 32.4 | 91.8 | 5.8 | 21.6 | 91.2 | 75.0 | 198.1 | 62.2 |
| STD ($\tau = 1$) | 58.6 | 11.4 | 57.2 | 41.3 | 42.2 | 33.6 | 43.1 | 75.2 | 108.2 | 31.0 | 30.3 | 85.4 | 5.7 | 20.4 | 85.6 | 70.9 | 168.0 | 57.0 |
| ACT | 126.7 | 20 | 124.8 | 79.9 | 82.1 | 66.3 | 124.2 | 147.7 | 191.6 | 55.4 | 61.2 | 180.1 | 7.4 | 33.8 | 150.8 | 123.9 | 377.8 | 114.9 |
| GLOB | 44.5 | 9.3 | 43.0 | 32.0 | 32.1 | 25.9 | 31.4 | 58.5 | 83.9 | 23.8 | 23.1 | 64.6 | **5.1** | 16.2 | 66.4 | 54.7 | 114.6 | 42.9 |
| TRIM | 43.3 | 8.8 | 41.8 | 31.2 | 30.7 | 24.6 | 29.4 | 56.2 | **82.3** | 23.4 | 22.6 | **62.7** | 5.1 | **16.0** | **64.1** | **53.2** | **99.0** | **40.8** |
| SPEC | 42.1 | **8.7** | 40.6 | 30.3 | 29.8 | 23.8 | 28.0 | 54.8 | 87.0 | 24.9 | 23.8 | 67.5 | 5.2 | 16.8 | 69.1 | 57.1 | 124.2 | 43.2 |
| Min Imp (%) | 24 | 19 | 25 | 23 | 24 | 23 | 27 | 22 | 20 | 20 | 21 | 21 | 8 | 18 | 19 | 20 | 26 | 24 |
| Max Imp (%) | 28 | 24 | 29 | 27 | 29 | 29 | 35 | 27 | 24 | 25 | 25 | 27 | 11 | 22 | 25 | 25 | 41 | 28 |

Table 13: Validation perplexity (↓) for our 12-block models trained on `The Pile` when performing continued pre-training starting from a pre-trained embedding matrix. DEPT performs worse than for the 24-block trained on `The Pile` and than for our $MC4$ models. However, when considering Table 12, we can observe it wins all comparisons by wide margins when starting from a randomly initialized embedding matrix, indicating that this gap is driven by the embedding space being fitted to the high-resource languages despite the baselines having a **worse** transformer body.

| Name (UNIGRAM-CE) | NH (8.1) | GH (7.9) | PA (8.2) | UB (7.8) | FL (7.8) | EE (7.9) | EP (10) | WK (8.2) | CC (7.9) | SE (7.7) | PC (8) | PP (9.1) | DM (6.9) | AX (7.7) | GU (7.7) | HN (7.7) | UI-OOD (10) | AVG (8.1) |
|---|---|---|---|---|---|---|---|---|---|---|---|---|---|---|---|---|---|---|
| STD ($\tau = 0$) | 31.9 | 7.6 | 30.5 | 24.2 | 23.2 | **18.1** | **20.5** | 41.2 | 64.6 | 18.9 | 19.2 | 49.4 | **4.9** | 13.6 | 51.7 | 42.6 | 68.4 | 31.2 |
| STD ($\tau = 1$) | 32.7 | **6.6** | 29.5 | 23.4 | **21.3** | 20.8 | 27.8 | **38.4** | **57.9** | **17.0** | **17.2** | 56.8 | 5.0 | **12.0** | **51.2** | 42.8 | 81.8 | 31.9 |
| GLOB | 30.2 | 7.1 | 29.8 | 22.9 | 23.7 | 20.0 | 21.9 | 41.6 | 61.8 | 17.9 | 19.6 | 48.0 | 5.2 | 13.7 | 54.1 | 42.2 | 90.7 | 32.4 |
| TRIM | **29.5** | 6.9 | **29.2** | **22.4** | 23.0 | 19.4 | 21.1 | 40.8 | 60.6 | 17.4 | 19.2 | **46.7** | 5.1 | 13.4 | 52.4 | **41.0** | **81.1** | **31.1** |
| Min Imp (%) | **5.4** | **−7.0** | −1.2 | 2.0 | −11.2 | −10.5 | −6.7 | −8.3 | −6.8 | −5.1 | −13.8 | **3.0** | −5.5 | −14.3 | −5.6 | **0.9** | −32.7 | −3.7 |
| Max Imp (%) | **7.5** | **−4.0** | 1.0 | 4.3 | −7.9 | −7.5 | −2.9 | −6.2 | −4.7 | −2.6 | −11.4 | **5.6** | −4.2 | −11.5 | −2.3 | **3.8** | −18.6 | **0.3** |

embeddings or leveraging pre-trained ones. This consistency underscores the robustness of DEPT's transformer body across varying embedding initialization and sampling strategies.

## B.7 SCALING EXPERIMENTS

Here, we train smaller multi-domain models with 12 blocks to validate the scaling properties of DEPT across model sizes. In Table 12, we observe that, similar to Table 10, DEPT models outperform all baselines significantly when starting from random initialization. Importantly, the embeddings constitute a larger percentage of the model parameters at this model scale. This highlights the robustness of DEPT's modifications to the embedding space in enabling the training of a better transformer body.

Interestingly, when using pre-trained embeddings (Table 6), the smaller DEPT models perform worse than their larger counterparts. We speculate that the amount of local per-source training performed by DEPT prior to OuterOpt should scale with model size. At this scale, the aggregation procedure may be overly harsh on the embedding parameters, particularly for the GLOB and TRIM configurations. This suggests that careful adjustments to the aggregation procedure may be necessary to maintain DEPT's effectiveness at smaller model scales.

## B.8 COMPARISON AGAINST SINGLE-CLIENT MODELS

To study the impact of model averaging, we now compare DEPT-based models with models trained on isolated data sources that are never averaged/merged. For fair comparisons, the model of each data source has seen as many tokens as it would have as a component in DEPT-based training and has undergone continued pre-training for the same number of steps with access to the full dataset. Additionally, DEPT models have undergone continued pre-training from random initialization. If we had compared against such models without the continued pre-training phase, they would have dominated on their respective data source while losing all other comparisons, especially in the case of multilingual data.

**Tables 14 and 15** show how DEPT models perform when all participants start from random initialization. Since the models trained on isolated data sources do not get to keep their highly specialized embeddings, this comparison evaluates how generalizable the abstractions learnt by the transformer

Table 14: Validation perplexity (↓) for 24-block models trained on `The Pile` after **continued pre-training** with **proportional** sampling from **randomly-initialized** embeddings, compared to models which had been pre-trained on a single data source for the same total number of tokens as DEPT has seen from their distributions. DEPT outperforms all baselines. DEPT outperforms all baselines. Baselines whose pre-training dataset matches the evaluation dataset are highlighted in olive.

| Name (UNIGRAM-CE) | DM (6.9) | EE (7.9) | EP (10) | FL (7.8) | GH (7.9) | CC (7.9) | PA (8.2) | SE (7.7) | PP (9.1) | WK (8.2) | AX (7.7) | UB (7.8) | PC (8) | NH (8.1) | GU (7.7) | HN (7.7) | UI-OOD (10) | AVG (8.1) |
|---|---|---|---|---|---|---|---|---|---|---|---|---|---|---|---|---|---|---|
| CC | 4.8 | 29.3 | 44.4 | 20 | 5.9 | 54.4 | 30 | 16.4 | 77.3 | 37.7 | 10.8 | 23.2 | 15.8 | 38 | 52 | 44.7 | 109.2 | 36.1 |
| PC | 4.8 | 28.1 | 40.3 | 19.1 | 5.7 | 52.1 | 27.9 | 15.7 | 72.8 | 35.8 | 10.4 | 21.9 | 14.8 | 35.5 | 50.3 | 43.4 | 110.6 | 34.7 |
| AX | 4.9 | 28.9 | 41.7 | 19.8 | 5.7 | 53.5 | 29.1 | 15.9 | 74.4 | 36.8 | 10.5 | 22.5 | 15.3 | 36.8 | 52.1 | 44.8 | 97.4 | 34.7 |
| GH | 4.9 | 30.1 | 43.6 | 20.8 | 5.8 | 55.9 | 30.7 | 16.5 | 78.2 | 38.5 | 10.9 | 23.8 | 16 | 38.9 | 54.5 | 46.7 | 120.7 | 37.4 |
| FL | 4.9 | 31.8 | 49.8 | 21.5 | 6.3 | 58.5 | 32.4 | 17.4 | 84.2 | 40.9 | 11.4 | 24.8 | 16.8 | 41 | 55.8 | 47.9 | 122.1 | 39.3 |
| SE | 4.8 | 28.2 | 42.2 | 19.4 | 5.6 | 52.9 | 29 | 15.5 | 75 | 36.6 | 10.5 | 22.4 | 15.3 | 36.8 | 50.8 | 43.4 | 100 | 34.6 |
| WK | 4.8 | 28.1 | 42.1 | 18.9 | 5.7 | 51.6 | 28.4 | 15.8 | 74 | 34.8 | 10.5 | 21.9 | 15.1 | 35.7 | 49.8 | 43.2 | 95.4 | 33.9 |
| DM | 7.3 | 140.4 | 559.4 | 100.6 | 28 | 239.5 | 184 | 71 | 543.9 | 213.9 | 34.8 | 121.7 | 69.3 | 213.9 | 193 | 170 | 966.8 | 226.9 |
| GLOB | 4.8 | 25.7 | 38.2 | 17.3 | 5.4 | 47.7 | 25.7 | 14.7 | 68.3 | 32.7 | 9.9 | 20 | 14 | 32.2 | 46.5 | 39.8 | 94.8 | 31.6 |
| TRIM | 4.8 | 27.3 | 39.5 | 18.5 | 5.6 | 51.2 | 27.8 | 15.4 | 71.8 | 35.1 | 10.3 | 21.7 | 14.8 | 35.1 | 49.1 | 42.2 | 95.7 | 33.3 |
| SPEC | 4.8 | 26.7 | 36.8 | 18.2 | 5.5 | 50.1 | 27.1 | 15.1 | 69.1 | 34.2 | 10.1 | 21.1 | 14.5 | 34.3 | 48.5 | 41.7 | 97.6 | 32.7 |
| SPEC-OPT | 4.7 | 25.9 | 35 | 17.5 | 5.4 | 48.3 | 26.1 | 14.7 | 66.6 | 32.8 | 9.9 | 20.4 | 14.1 | 32.9 | 47.3 | 40.5 | 88.6 | 31.2 |
| Min Imp (%) | 0.7 | 2.7 | 2 | 1.6 | 0.5 | 0.8 | 0.5 | 0.3 | 1.4 | −0.9 | 1.2 | 1 | 0 | 1 | 1.3 | 2.1 | −2.3 | 1.7 |
| Max Imp (%) | 1.2 | 8.3 | 13.2 | 8.4 | 4.3 | 7.6 | 7.9 | 5.3 | 8.5 | 6 | 5 | 8.8 | 5.6 | 9.2 | 6.5 | 7.8 | 7.1 | 7.8 |

Table 15: Validation perplexity (↓) for 12-block models trained on `MC4` after **continued pre-training** with **unfiorm** sampling from **randomly-initialized** embeddings, compared to models which had been pre-trained on a single data source for the same total number of tokens as DEPT has seen from their distributions. Baselines whose pre-training dataset matches the evaluation dataset are highlighted in olive.

| | In-Distribution | | | | | | | | | Out-of-Distribution | | | |
|---|---|---|---|---|---|---|---|---|---|---|---|---|---|
| | In-Distribution | | | | | | | | | Out-of-Distribution | | | |
| Name (UNIGRAM-CE) | ZH (9.8) | UR (10.5) | MS (9.2) | IT (7.7) | SR (10.5) | LA (9) | EN (7.5) | SW (10) | Avg (In-D) (9.3) | EL (14.4) | HI (13.9) | DE (9.7) | Avg (OOD) (12.6) |
| ZH | 187.8 | 44.6 | 113.9 | 98.6 | 89.1 | 73.9 | 128.8 | 76 | 101.6 | 5744.8 | 6476.5 | 1448 | 4556.4 |
| UR | 94.8 | 27.5 | 66 | 56.9 | 48.7 | 42.5 | 79.3 | 44.1 | 57.5 | 2596.5 | 2371.1 | 690.5 | 1886 |
| MS | 78.8 | 24.8 | 58 | 50 | 42.7 | 37.3 | 70 | 38.9 | 50.1 | 2673.3 | 2329.2 | 599.4 | 1867.3 |
| IT | 81.3 | 25.1 | 59.7 | 51 | 43.8 | 38.2 | 71.8 | 39.8 | 51.3 | 2617.2 | 2256.9 | 615.3 | 1829.8 |
| SR | 85.4 | 25.7 | 61 | 52.4 | 44.9 | 39.4 | 73.7 | 40.9 | 52.9 | 2992.4 | 2648.3 | 657.2 | 2099.3 |
| LA | 104.8 | 29.6 | 71.7 | 60.6 | 53.5 | 46 | 85.2 | 47.6 | 62.4 | 2838.7 | 2824.8 | 746.6 | 2136.7 |
| EN | 104.4 | 30.1 | 71.9 | 61.2 | 54.3 | 46.2 | 85 | 47.8 | 62.6 | 3344.4 | 3360.6 | 834.8 | 2513.3 |
| SW | 79.3 | 24.7 | 58.1 | 49.9 | 43 | 37.3 | 69.9 | 39 | 50.1 | 2552.3 | 2067.5 | 608.3 | 1742.7 |
| GLOB | 67.7 | 22.4 | 53.7 | 46 | 38.6 | 33.9 | 65.4 | 35.2 | 45.4 | 2308.3 | 1676.5 | 559.5 | 1514.7 |
| TRIM | 67.7 | 22.8 | 55.2 | 47.5 | 39.7 | 35.1 | 67.2 | 36.3 | 46.4 | 2547.7 | 1911 | 567.4 | 1675.4 |
| SPEC | 69.5 | 23 | 55.4 | 47.8 | 40.3 | 34.7 | 68.1 | 36.3 | 46.9 | 2232.1 | 1578.8 | 544.7 | 1451.9 |
| Min Imp (%) | 11.8 | 6.6 | 4.4 | 4.2 | 5.7 | 6 | 2.7 | 6.8 | 6.4 | 0.2 | 7.6 | 5.3 | 3.9 |
| Max Imp (%) | 14.1 | 9.3 | 7.4 | 7.7 | 9.6 | 9.1 | 6.5 | 9.6 | 9.4 | 12.5 | 23.6 | 9.1 | 16.7 |

body are across datasets. In the case of `The Pile`, shown in Table 14, DEPT models outperform the isolated baselines by 7.8% in terms of average perplexity. Crucially, DEPT models win all comparisons even though isolated baselines get evaluated on their pre-training dataset, indicating that they have not learned superior abstractions even in this case. For `MC4`, Table 15 show a very similar trend with a much higher degree of outperformance for DEPT, 9.8% on average on in-distribution data and 16.7% for out-of-distribution (OOD) data, likely because the transformer body learned for one language has significant difficulty in adapting to a multilingual context.

**Tables 16 and 17** show the impact of keeping the pre-trained embeddings before continued pre-training. The impact of this change is as expected: embeddings pre-trained on a specific dataset perform well on that dataset; however, they fail to generalize. In the case of `The Pile`, shown in Table 16, DEPT loses most comparisons to the baseline trained on a given dataset; however, it outperforms in terms of average perplexity by a remarkable 27%. For `MC4`, shown in Table 17, the outperformance in terms of average perplexity is even more significant, 30.6% for in-distribution data and 14.9% of OOD data.

Table 16: Validation perplexity (↓) for 24-block models trained on `The Pile` after **continued pre-training** with **proportional** sampling from **pre-trained** embeddings, compared to models which had been pre-trained on a single data source for the same total number of tokens as DEPT has seen from their distributions. DEPT significantly outperforms in terms of average perplexity but gets beaten by specialized models on their respective data source. Baselines whose pre-training dataset matches the evaluation dataset are highlighted in olive.

| Name (UNIGRAM-CE) | DM (6.9) | EE (7.9) | EP (10) | FL (7.8) | GH (7.9) | CC (7.9) | PA (8.2) | SE (7.7) | PP (9.1) | WK (8.2) | AX (7.7) | UB (7.8) | PC (8) | NH (8.1) | GU (7.7) | HN (7.7) | UI-OOD (10) | AVG (8.1) |
|---|---|---|---|---|---|---|---|---|---|---|---|---|---|---|---|---|---|---|
| CC | 19.6 | 57.6 | 294.4 | 35.1 | 32.2 | 30.8 | 41.1 | 47.9 | 133.1 | 35.8 | 44.2 | 26 | 47 | 43.7 | 52.8 | 38.2 | 79.2 | 62.3 |
| PC | 4.8 | 25.5 | 37.5 | 17.3 | 5.5 | 45.5 | 17.7 | 14.8 | 63 | 30.7 | 9.4 | 17.6 | 10.1 | 23.2 | 46.4 | 39 | 100.8 | 29.9 |
| AX | 4.8 | 27 | 38.5 | 18.7 | 5.5 | 49.7 | 25.9 | 14.7 | 64.8 | 33.8 | 7.4 | 19.5 | 13.8 | 32.9 | 49.5 | 41.4 | 110.3 | 32.8 |
| GH | 5 | 31.1 | 47.3 | 24.4 | 3.9 | 62.6 | 37.1 | 13.3 | 80.6 | 44.5 | 11.6 | 26.4 | 17.9 | 47.6 | 60.8 | 47.1 | 70.1 | 37.1 |
| FL | 4.9 | 23.8 | 49.6 | 10.1 | 5.9 | 45 | 27.3 | 15.8 | 73.4 | 31.6 | 10.6 | 20.4 | 14.7 | 33.6 | 42.5 | 34.8 | 99.5 | 32.2 |
| SE | 4.7 | 24.3 | 38.3 | 17.9 | 4.4 | 45.7 | 27.1 | 9.3 | 62.1 | 33 | 9.5 | 19.8 | 14.3 | 34.3 | 46.1 | 34.7 | 70.2 | 29.2 |
| WK | 4.9 | 23.8 | 33.8 | 16.4 | 5.7 | 40 | 25.1 | 15.2 | 57.4 | 18.6 | 10.2 | 19.4 | 13.9 | 31.3 | 39.5 | 37.4 | 98.7 | 28.9 |
| DM | 4.4 | 81.6 | 210.9 | 59.3 | 14.4 | 143.8 | 99.7 | 41.2 | 248.4 | 116.5 | 22.6 | 67.9 | 40.8 | 124.1 | 126.1 | 108.1 | 424.8 | 113.8 |
| GLOB | 4.5 | 17 | 16.1 | 13.2 | 4.5 | 34.5 | 17.9 | 11.2 | 37.8 | 22.4 | 8.4 | 14.4 | 11 | 20.6 | 35.5 | 28.3 | 61.2 | 21.1 |
| TRIM | 4.6 | 20.5 | 23 | 13.9 | 4.6 | 38 | 20.2 | 12 | 49.9 | 25.1 | 8.7 | 16.6 | 11.8 | 25.7 | 38 | 32.9 | 56.8 | 23.7 |
| Min Imp (%) | −2.9 | 13.9 | 32 | −37.4 | −19.8 | −23.5 | −14.1 | −28.2 | 13.1 | −34.8 | −18.7 | 5.5 | −16.9 | −10.9 | 3.8 | 5.3 | 12.7 | 18.1 |
| Max Imp (%) | −1 | 28.5 | 52.3 | −31.1 | −16.6 | −12 | −1.1 | −19.8 | 34.2 | −20.4 | −14.3 | 18.1 | −9.3 | 11.2 | 10.1 | 18.5 | 19 | 27 |

Table 17: Validation perplexity (↓) for 12-block models trained on `MC4` after **continued pre-training** with **uniform** sampling from **pre-trained** embeddings, compared to models which had been pre-trained on a single data source for the same total number of tokens as DEPT has seen from their distributions. DEPT significantly outperforms in terms of average perplexity but gets beaten by specialized models on their respective data source. Baselines whose pre-training dataset matches the evaluation dataset are highlighted in olive.

| | In-Distribution | | | | | | | | | Out-of-Distribution | | | |
|---|---|---|---|---|---|---|---|---|---|---|---|---|---|
| Name (UNIGRAM-CE) | ZH (9.8) | UR (10.5) | MS (9.2) | IT (7.7) | SR (10.5) | LA (9) | EN (7.5) | SW (10) | Avg (In-D) (9.3) | EL (14.4) | HI (13.9) | DE (9.7) | Avg (OOD) (12.6) |
| ZH | 33.3 | 36.6 | 87.1 | 71.3 | 68.7 | 55.9 | 90.2 | 58.1 | 62.6 | 5351.5127 | 3197.97314 | 936.293457 | 3161.92643 |
| UR | 124.7 | 12.7 | 70.8 | 62.9 | 57.4 | 49.5 | 76.8 | 49.8 | 63.1 | 3189.1106 | 774.645325 | 802.290588 | 1588.68217 |
| MS | 89.5 | 26 | 27.8 | 48.1 | 47.1 | 38.2 | 58.9 | 38.3 | 46.7 | 2983.83643 | 2491.64209 | 620.154907 | 2031.87781 |
| IT | 91.4 | 27.4 | 59.6 | 24.4 | 46.9 | 35.8 | 59.6 | 40.7 | 48.2 | 2353.07642 | 3057.45215 | 344.762726 | 1918.43043 |
| SR | 99.3 | 28.1 | 65.1 | 52.5 | 20.2 | 41.7 | 70.1 | 43.6 | 52.6 | 2174.83203 | 3398.59985 | 643.610352 | 2072.34741 |
| LA | 100.2 | 30.2 | 66.9 | 48.1 | 51 | 20.1 | 68.1 | 45.1 | 53.7 | 716.628967 | 2479.87817 | 240.702454 | 1145.73653 |
| EN | 1142.1 | 150.7 | 276.5 | 211.1 | 408.8 | 147.1 | 86.2 | 169.1 | 323.9 | 1315877.75 | 679658.688 | 11445.9727 | 668994.137 |
| SW | 91.6 | 26.2 | 55.9 | 48.3 | 47 | 38.1 | 58.3 | 17.8 | 47.9 | 2782.6792 | 2673.07813 | 557.471863 | 2004.40973 |
| GLOB | 40.1 | 15.5 | 30.1 | 39.6 | 39 | 29.7 | 40.5 | 24.6 | 32.4 | 1737.3 | 823.4 | 335.1 | 965.3 |
| TRIM | 41.9 | 16.2 | 31.3 | 41.3 | 40.8 | 30.8 | 42 | 25.6 | 33.7 | 1725 | 855.2 | 345.6 | 975.3 |
| Min Imp (%) | −26.1 | −28.1 | −12.8 | −68.9 | −101.9 | −53 | 28 | −43.9 | 27.8 | −142.4 | −10.4 | −43.6 | 14.9 |
| Max Imp (%) | −20.6 | −22.8 | −8.5 | −62.1 | −92.7 | −47.4 | 30.6 | −38.6 | 30.7 | −142.4 | −10.4 | −43.6 | 14.9 |

## B.9 COMPARISON AGAINST PYTHIA

The experiments in our work are designed to investigate the outlined research questions instead of producing a state-of-the-art model. However, we believe that providing a comparison against a standard baseline may help better contextualize the performance of a given DEPT model. For this purpose, we chose `Pythia` (Biderman et al., 2023) as it shares a very similar architecture to the one used in our work, with the only exception being that `Pythia` uses untied weights for the embedding matrix and thus all its equivalent sizes have more model parameters. `Pythia` models are trained on one epoch of the entire `The Pile`, 300B tokens, regardless of size. Thus, we do not have any OOD dataset for them, and they are expected to perform better on `Ubuntu IRC`. Since they are trained for many more tokens than DEPT models, we do not perform additional continued pre-training when starting from a pre-trained embedding matrix (the extra tokens existed to equalize the amount of work done across baselines); thus, those comparisons show the raw performance of `Pythia` as published by its authors. When starting from a random initialization we use the standard procedure from above.

**Table 18** shows a comparison between a 160M `Pythia` model and the 125M DEPT models when starting from pre-trained embeddings. At this scale, the additional pre-training of `Pythia` (using $30\times$ the tokens of DEPT) does not provide an evident advantage as the model capacity is insufficient to benefit from it. Thus, outside of the expected outperformance on `Ubuntu IRC (UI)`, `Pythia-160M` performs similarly to DEPT models and is slightly outperformed on average. We also speculate that using the full 22-dataset version of `The Pile` during pre-training likely reduced the performance of `Pythia-160M` as it had to fit a broader data distribution. We do not provide

Table 18: Validation perplexity (↓) for our 12-block models trained on `The Pile` when performing continued pre-training using **uniform** sampling starting from a **pre-trained** embedding matrix. `DEPT` slightly outperforms `Pythia-160M` at this small scale as its $30\times$ greater number of tokens is not beneficial with insufficient model capacity. `Pythia-160M` was trained on `Ubuntu IRC (UI)`, thus its outperformance is expected as it is not an OOD dataset for this model.

| Name (UNIGRAM-CE) | NH (8.1) | GH (7.9) | PA (8.2) | UB (7.8) | FL (7.8) | EE (7.9) | EP (10) | WK (8.2) | CC (7.9) | SE (7.7) | PC (8) | PP (9.1) | DM (6.9) | AX (7.7) | GU (7.7) | HN (7.7) | UI-OOD (10) | AVG (8.1) |
|---|---|---|---|---|---|---|---|---|---|---|---|---|---|---|---|---|---|---|
| PYTHIA-160M | 47.3 | 8.2 | 36.8 | 31.4 | 24.1 | 34 | 32.8 | **40.2** | 64.1 | 22.4 | 21.35 | 74.5 | 6.8 | 16.3 | 55 | 54.9 | **24.31** | 33.1 |
| GLOB | 30.2 | 7.1 | 29.8 | 22.9 | 23.7 | 20.0 | 21.9 | 41.6 | 61.8 | 17.9 | 19.6 | 48.0 | 5.2 | 13.7 | 54.1 | 42.2 | 90.7 | 32.4 |
| TRIM | **29.5** | **6.9** | **29.2** | **22.4** | **23.0** | **19.4** | **21.1** | 40.8 | **60.6** | **17.4** | **19.2** | **46.7** | **5.1** | **13.4** | **52.4** | **41.0** | 81.1 | **31.1** |

random initialization results for this model size since we found it impossible to make it behave well during continued pre-training, and we believe the comparison would be unfair.

Table 19: Validation perplexity (↓) for 24-block models trained on `The Pile` after **continued pre-training** with **proportional** sampling from **randomly-initialized** embeddings, compared to `Pythia-410M`. DEPT models come close to `Pythia-410M` despite the latter being trained on $10\times$ more tokens, indicating a comparable if slightly worse transformer body. `Pythia-410M` was trained on `Ubuntu IRC (UI)`, thus its outperformance is expected as it is not an OOD dataset for this model.

| Name (UNIGRAM-CE) | DM (6.9) | EE (7.9) | EP (10) | FL (7.8) | GH (7.9) | CC (7.9) | PA (8.2) | SE (7.7) | PP (9.1) | WK (8.2) | AX (7.7) | UB (7.8) | PC (8) | NH (8.1) | GU (7.7) | HN (7.7) | UI-OOD (10) | AVG (8.1) |
|---|---|---|---|---|---|---|---|---|---|---|---|---|---|---|---|---|---|---|
| PYTHIA-410M | 4.9 | 25.9 | 43.3 | 17.4 | **5.1** | **45.6** | **24.7** | **13.9** | 65.7 | **31.7** | 9.6 | **18.8** | **13.5** | 31.3 | 44.5 | 38.3 | 81.2 | **30.3** |
| GLOB | 4.8 | **25.7** | 38.2 | **17.3** | 5.4 | 47.7 | 25.7 | 14.7 | 68.3 | 32.7 | 9.9 | 20 | 14 | 32.2 | 46.5 | 39.8 | 94.8 | 31.6 |
| TRIM | 4.8 | 27.3 | 39.5 | 18.5 | 5.6 | 51.2 | 27.8 | 15.4 | 71.8 | 35.1 | 10.3 | 21.7 | 14.8 | 35.1 | 47.3 | 42.2 | 95.7 | 33.3 |
| SPEC | 4.8 | 26.7 | 36.8 | 18.2 | 5.5 | 50.1 | 27.1 | 15.1 | 69.1 | 34.2 | 10.1 | 21.1 | 14.5 | 34.3 | 48.5 | 41.7 | 97.6 | 32.7 |
| SPEC-OPT | **4.7** | 25.9 | **35** | 17.5 | 5.4 | 48.3 | 26.1 | 14.7 | 66.6 | 32.8 | 9.9 | 20.4 | 14.1 | 32.9 | 47.3 | 40.5 | 88.6 | 31.2 |

Table 20: Validation perplexity (↓) for 24-block models trained on `The Pile` after **continued pre-training** with **proportional** sampling from **randomly-initialized** embeddings, compared to `Pythia-410M`. `Pythia-410M` significantly outperforms `DEPT` as its $30\times$ larger number of training tokens allow it to train much better embeddings. `Pythia-410M` was trained on `Ubuntu IRC (UI)`, thus its outperformance is expected as it is not an OOD dataset for this model.

| Name (UNIGRAM-CE) | DM (6.9) | EE (7.9) | EP (10) | FL (7.8) | GH (7.9) | CC (7.9) | PA (8.2) | SE (7.7) | PP (9.1) | WK (8.2) | AX (7.7) | UB (7.8) | PC (8) | NH (8.1) | GU (7.7) | HN (7.7) | UI-OOD (10) | AVG (8.1) |
|---|---|---|---|---|---|---|---|---|---|---|---|---|---|---|---|---|---|---|
| PYTHIA-410M | 3.8 | 9.7 | 8.9 | 7.7 | 3 | 19 | 11.8 | 7.2 | 21.3 | 12.5 | 5.9 | 10.3 | 7.6 | 15 | 17.6 | 16.9 | 7.8 | 10.9 |
| GLOB | 4.5 | 17 | 16.1 | 13.2 | 4.5 | 34.5 | 17.9 | 11.2 | 37.8 | 22.4 | 8.4 | 14.4 | 11 | 20.6 | 35.5 | 28.3 | 61.2 | 21.1 |
| TRIM | 4.6 | 20.5 | 23 | 13.9 | 4.6 | 38 | 20.2 | 12 | 49.9 | 25.1 | 8.7 | 16.6 | 11.8 | 25.7 | 38 | 32.9 | 56.8 | 23.7 |

**Tables 19 and 20** show the expected outperformance of the 410M `Pythia` model over the `DEPT` models, as this size has sufficient capacity to benefit from the extensive ($10\times$ longer compared to `DEPT`) pre-training. When starting from a random initialization, Table 19, the best `DEPT` variant is within 1 average perplexity point of `Pythia-410M`, indicating that a large portion of the additional token budget is primarily used to obtain better embeddings without providing a significantly improved transformer body. When starting from pre-initialized embeddings, Table 20, `Pythia-410M` significantly outperforms `DEPT` achieving an average perplexity 10 points lower than the best `DEPT` variant. As discussed above, this is driven by its more extensive pre-training and improved embeddings.

# C APPLICATIONS

## C.1 FEDERATED PRE-TRAINING OF LLMS ON MULTILINGUAL POPULATION

The challenges of training under data heterogeneity have come back into focus with recent forays into federated pre-training (Douillard et al., 2023; Sani et al., 2024; Charles et al., 2023; Nous Research, 2024), triggered in equal parts by privacy concerns, compute sharing and the search for more data in previously untapped reservoirs.

The way in which datasets are curated, filtered, and combined has a significant impact on the performance of LLMs (Long et al., 2024). Determining the best methods for data curation, filtering, and mixing from various sources requires extensive experimentation to identify configurations that optimize performance on target evaluation metrics (Meta, 2024). Consequently, the specific details of these processes are often closely guarded by leading LLM developers. Despite careful dataset preparation, data heterogeneity remains inevitable due to the inherent imbalance in data sources. One of the most prominent imbalances is in language representation. For instance, only about 5% of the pre-training data for Llama3 is non-English, covering over 30 languages, which results in lower expected performance in non-English contexts (Meta, 2024). A similar performance disparity across languages has also been observed with GPT-4 (Achiam et al., 2023).

Current datasets used for pre-training are highly geographically concentrated to a few areas of the globe (Faisal et al., 2022), providing the so-called high-resource languages, with high-quality domain-specific data being available predominantly in such languages (Magueresse et al., 2020). Such datasets are collected from internet sources and then curated (Brown et al., 2020; Dubey et al., 2024). However, bottlenecks in the rate of high-quality data generation (Villalobos et al., 2022) and copyright concerns (Grynbaum & Mac, 2023) have led to large organizations making deals with private data providers such as publishers (OpenAI, 2023; Patel & Palazzolo, 2024) in order to meet the demand of ever-growing models.

Federated pre-training as a methodology allows the model to be taken directly to the training data, potentially enabling training under privacy concerns or legislation that limits data movement (Woisetschläger et al., 2024). While this has obvious applications for collaborative training of LMs, it can also be applied by a single organization as a drop-in replacement for mini-batch SGD during pre-training (Douillard et al., 2023), which eliminates dataset movement while massively lowering the communication frequency of model training compared to Data-parallel algorithms (Rajbhandari et al., 2020) which need to synchronize gradients every batch. The version of the algorithm used in a centralized setting, mathematically equivalent to Federated Averaging (McMahan et al., 2017a), has alternatively been known as: (a) communication-efficient SGD (Yu et al., 2019), (b) Local SGD (Stich, 2019; Ortiz et al., 2021), or (c) as a specific variant of the REPTILE (Nichol et al., 2018) meta-learning algorithm. Under these various methodologies, it has been shown to (a) confer a linear speedup to convergence similar to increasing batch size, (b) provide better generalization to models compared to standard large-batch training, (c) enable meta-learning across various tasks.

While the current centralized pre-training recipe may be stabilized with great effort, such measures are largely impractical in federated training scenarios where the participants refuse to offer full control over their data to a third party or where the underlying training distribution may shift as new participants enter a federation or old ones exit. Furthermore, the complexity of the current pipeline is impractical to all except the best-funded organizations, even in a centralized training context.

The inability to directly inspect data sources in a federated context makes it impossible to construct a dedicated vocabulary for a data mixture, ensure a standard curation pipeline on a per-sample basis, or strongly control data sampling rates across all sources. Motivated by this extreme setting, we aim to construct a pre-training procedure that is capable of learning from multiple highly heterogeneous data sources without model divergence while providing a foundation model with greater **generalization** and more **plasticity** in adapting to new data.

## D  TRAINING UNDER DATA HETEROGENEITY

Training LLMs such as Llama 3 (Dubey et al., 2024) requires extensive manual tuning, heuristics, and model-based data selection procedures. This effort aims to achieve the desired mix of categories, such as general knowledge, mathematics, coding, and multilingual data.

This complexity arises due to the wide range of capabilities required by LMs and the risk of negative interference across domains and languages. Current pre-training methodologies are prone to divergence unless data sampling ratios can be meticulously curated based on the characteristics of the data and its fit to the model's distribution at any given time (Dubey et al., 2024). Multi-domain ratios are manually curated for downstream performance, requiring extensive and expensive tuning, while multilingual pre-training often employs temperature-weighted sampling (Devlin et al., 2019; Conneau et al., 2020; Xue et al., 2021) due to the vast number of languages involved,

As illustrated in Fig. 2, pre-training on heterogeneous data can result in model activation divergence (Hoffmann et al., 2022), even with a sampling temperature of $\tau = 1.0$, which corresponds to proportional sampling based on dataset size. Activation divergence is a precursor to significant, often irrecoverable, increases in loss, and necessitating model re-starts from earlier checkpoints with lower learning rates (Zhang et al., 2022). Longer training durations could be achieved by disproportionately sampling from larger, lower-quality datasets like `C4` or high-resource languages like English in multilingual pre-training. Alternatively, methods like active forgetting via embedding resetting (Chen et al., 2023), `ACT`, may artificially extend the training duration past the natural divergence point.

Previous studies show that this *Curse of Multilinguality* and/or *Negative Interference* can be attributed to vocabulary dilution and capacity contention (Conneau et al., 2020), language-specific parameter emergence (Wang et al., 2020), and suboptimal tokenization (Rust et al., 2021). Increasing model and vocabulary size helps capacity contention (Conneau et al., 2020; Wang et al., 2020), but this requires immense hardware resources (Dubey et al., 2024) to shard the model across multiple GPUS. Addressing vocabulary dilution in highly multilingual models is even more challenging, as providing enough tokens for all languages would result in impractically large models (Rust et al., 2021). These limitations drive us to find scalable methods to incorporate broader data mixtures without significantly increasing the in-memory model size during training.

## E    FINE-TUNING DEPT MODELS

We evaluate fine-tuning performance on three downstream tasks: RACE, MNLI, and STSB. All models are fine-tuned using the recipes provided by Radford et al. (2018) for each task using the AdamW optimizer with a linear learning rate scheduler. For RACE, the model is trained for 5 epochs with a learning rate of 6e-5 and a batch size of 16. For MNLI, fine-tuning is performed over 2 epochs with a learning rate of 4e-5 and a batch size of 32. Finally, STSB is fine-tuned for 5 epochs using a learning rate of 2e-5 and a batch size of 32. The results are reported in Table 21.

Table 21: The performance on downstream tasks (↑), following continued pre-training, shows that DEPT models achieve $3\% - 7.5\%$ relative improvements over the baselines, with TRIM delivering the best results. DEPT consistently outperforms baselines, even with pre-trained embedding initialization, underscoring the importance of an effective transformer body.

| Name | Random Init | | | | Pre-trained Init | | | |
|---|---|---|---|---|---|---|---|---|
| | RACE (ACC) | MNLI (ACC) | STSB (PC) | SST2 (ACC) | RACE (ACC) | MNLI (ACC) | STSB (PC) | SST2 (ACC) |
| **STD** ($\tau = 0$) | 0.5 | 0.6 | 0.66 | 0.79 | 0.5 | 0.71 | 0.74 | 0.81 |
| **STD** ($\tau = 1$) | 0.46 | 0.68 | 0.73 | 0.81 | 0.53 | 0.7 | 0.76 | 0.83 |
| **ACT** | 0.45 | 0.66 | 0.73 | 0.8 | – | – | – | – |
| **GLOB** | 0.51 | **0.72** | 0.78 | 0.83 | 0.51 | 0.69 | 0.76 | 0.82 |
| **TRIM** | **0.53** | 0.71 | 0.78 | 0.83 | **0.55** | **0.73** | **0.81** | **0.86** |
| **SPEC** | 0.52 | 0.71 | **0.79** | 0.81 | – | – | – | – |
| **SPEC-OPT** | 0.51 | 0.69 | 0.77 | **0.85** | – | – | – | – |
| **Min Imp (%)** | **2.9%** | **4.6%** | **5.9%** | $-0.7\%$ | $-3.7\%$ | $-3.2\%$ | **0.5%** | $-1.8\%$ |
| **Max Imp (%)** | **5.8%** | **6.1%** | **7.5%** | **4.1%** | **3.2%** | **3%** | **6.6%** | **3.2%** |

## F    USING SPEC MODELS FOR INFERENCE

As discussed in Sections 2.4, 3.5 and 6.1, SPEC models do not inherently support inference on a broad corpus after initial pre-training. Suppose local vocabularies and embedding matrices are available without privacy concerns. In that case, inference can be performed using the embedding matrix of the broadest data source or the one closest to the target application. For instance, targeting English text would utilize EN embeddings for MC4 or CC embeddings for `The Pile`. While effective, this limits generalization beyond the broadest dataset in the pre-training distribution.

To handle a corpus resembling a mixture of all pre-training data sources or unseen ones, SPEC models require a broader embedding matrix for good performance. This can be achieved through multi-phased adaptive or continued pre-training, starting with a random embedding matrix or the

broadest pre-training one, as demonstrated in this work. Alternatives include vocabulary/embedding transfer (Remy et al., 2024) or vocabulary matching (Xu et al., 2024). If these methods fail to reach the desired performance, additional optimization may be necessary to align the embeddings with the transformer body.

