# OpenReview forum: "DEPT: Decoupled Embeddings for Pre-training Language Models"
_ICLR.cc/2025/Conference — ICLR 2025 Oral_

### Official Review · Reviewer_xZ5j · 2024-10-16

**Soundness:** 3
**Presentation:** 3
**Contribution:** 4
**Rating:** 8
**Confidence:** 4

**Summary:**

This paper proposes a pre-training framework DEPT which allows the model to train without being bound to a shared global vocabulary. Three variants are introduced: Glob, Trim, and Spec. Although each pipeline has the same setting for the transformer block, the difference lies in how they deal with the embeddings. Among the three, Glob is close to the standard pre-training: a single global embedding matrix is used. Trim keeps a local embedding matrix for each data source, and each token in the matrix is also contained in the global vocabulary. With the federated learning framework, the updates of a specific token are then aggregated to the same token in the global embedding matrix. Spec is a fully decoupled version where there is a non-sharing local embedding matrix for each data source. The authors evaluate the DEPT by investigating the efficiency, generalization as well as plasticity.

**Strengths:**

- The paper is well-written and easy to follow.

- The idea of decoupling embedding matrix and transformer block in pre-training within the federated learning framework is novel.

- The authors answer the raised research questions with meaningful and extensive experiments.

- The results generally confirm that DEPT can improve the generalization and plasticity of the models.

**Weaknesses:**

- The data sources are not always clear given a dataset. The proposed pipeline only works if the domains are known. Otherwise, some manual or automatic clustering has to be used to create different sets of data.

- The multi-domain data is almost only in English. But for the multilingual data, the data of each language should also contain various domains. Therefore there are confounding variables. A natural question would be whether the model can generalize to the same domains across different languages.

- No downstream tasks in natural language understanding or generation are evaluated on the resulting models. But such further evaluation is important.

**Questions:**

- abstract: I believe "negative interference" and "curse of multilingual" are not parallel concepts. Curse of multilingual is a special type of interference because of the training data containing many languages.

- I don't find the authors clearly stating the "performance metrics" in Tables 1, 2, 3, 4.

- The authors are suggested to have aggregated statistics for Tables 1, 2, 3, and 4 (e.g., average) so that the authors can have a better understanding of which model generally performs the best. Additionally, the authors can bold the best number in each column.

- line 278: 50,257 instead of 50257.0

- Line 341 (b) tn the ?

---

> ### Author Response · Authors · 2024-11-16
> **Response to Reviewer xZ5j**
>
> # Dear Reviewer **xZ5j**,
>
> We express our gratitude for the valuable feedback of their review and for acknowledging our proposal's **novelty**, **quality**, and **soundness**. Please see our detailed responses below:
>
> - **RE: “Clarification on the data sources”**
>   We thank the reviewer for raising this interesting point. One of our assumptions is, indeed, that data sources are partitioned into different domains or languages, as they are for **The Pile [1]** and **MC4 [4]** datasets that we use. We believe that this is not a strong assumption.
>   - First, in many scenarios, this assumption holds: in **Federated Learning** settings, in **Multilingual datasets**, and **Multi-domain datasets**, the partitions are already identifiable without additional manual or automatic clustering.
>   - Second, our experimental results show that, in the case of **IID data** shown in **Section B.3**, DEPT performs no worse than standard pre-training.
>
> - **RE: “Investigating multiple domains for multiple languages simultaneously”**
>   This particular context pointed out by the Reviewer represents a very interesting setting for DEPT.
>   - Our experimental space is indeed limited to investigating **multiple domains** and **multiple languages** separately. This limitation arises from, to the best of our knowledge, the lack of open-source datasets allowing for such experimentation.
>   - The construction of such datasets, which would require gathering in-the-wild multi-domain data in multiple languages, needs careful consideration and extensive efforts to make such text corpora usable.
>   - We expect our method to generalize to the **same domain across different languages**. However, we would be very excited to evaluate DEPT on open-source datasets of this kind if the Reviewer can kindly point them to us.
>
> - **RE: “Evaluating the downstream performance of DEPT models”**
>   Thanks to the Reviewer's comment, we acknowledge the importance of such an evaluation and provide an additional evaluation in the revision we posted.
>   - We invite the Reviewer to kindly inspect the new revision, especially **Table 7**, where they can find such an assessment.
>   - We evaluate the natural language understanding tasks: **Natural Language Inference (MNLI)**, **Question Answering (RACE)**, and **Sentence Similarity (STSB)** and show that DEPT wins all of the comparisons, bringing improvements in **accuracy (RACE, MNLI)** and **Pearson correlation (STSB)**, with relative improvements of **3%-7.5% over the baselines**.
>   - For such comparisons, **TRIM** performs best with **GLOB** and **SPEC** on equal footing, proving further advantages to our **novel embedding decoupling methods** beyond memory and communication efficiency and language modeling performance.
>   - If the Reviewer has any other recommendations for downstream tasks, we are happy to incorporate them.
>
> - **RE: “Negative interference and curse of multilinguality not being parallel concepts”**
>   We acknowledge that **negative interference** is a superset of the **curse of multilinguality** and encompasses a much broader set of issues.
>   - The parallels we draw between the two are based on the work of [3] and are meant simply to connect why our methods are aimed at both **multi-domain** and **multilingual data**, despite highly heterogeneous subsets such as mathematics not being generally perceived as a fully different language.
>   - Our new abstract now states that **heterogeneous data sources suffer from “challenges such as” negative interference or the curse of multilinguality** to avoid making them seem equivalent.
>
> - **RE: “Identifying performance metrics, adding aggregated statistics, and bolding the best number in each column”**
>   We have fixed **table formatting issues**, clearly identified every performance metric, and made the **paper presentation consistent**.
>
> - **Thank you so much for catching the typos.**
>   We have fixed them (Line 246, Line 279 in the original submission).
>
> ---
>
> We are firmly committed to improving our submission, having uploaded a new revision that addresses most of the Reviewer's comments, and we are in the process of completing the additional evaluations we have been kindly asked to produce.
>
> We intend to progressively improve our work, and we kindly ask the Reviewer to **raise their scores** if they believe our rebuttal addresses their concerns in light of our **additional downstream task results**, **clarifications**, **revisions**, and **improved presentation**.
>
> ### References
>
> [1] Gao et al., *The pile: An 800GB dataset of diverse text for language modeling*.
> [2] Sani et al., *The future of large language model pre-training is federated*.
> [3] Wang et al., *On Negative Interference in Multilingual Models: Findings and A Meta-Learning Treatment*.
> [4] Xue et al., *mT5: A massively multilingual pre-trained text-to-text transformer*.
>
> **Kind regards,**
> *The Authors*

---

> > ### Comment · Reviewer_xZ5j · 2024-11-18
> >
> > Dear authors,
> >
> > Thank you for your response. Most of my concerns and questions have been addressed/answered. It's a good paper worth accepting and I increased the score to 8. Good luck!

---

> > > ### Author Response · Authors · 2024-11-18
> > >
> > > Dear Reviewer xZ5j,
> > >
> > > Thank you for your thoughtful feedback and for taking the time to review and address our work. We greatly appreciate your kind words and the updated score. Your insights have been invaluable in refining our paper.

---

### Official Review · Reviewer_vuvz · 2024-10-28

**Soundness:** 4
**Presentation:** 2
**Contribution:** 4
**Rating:** 8
**Confidence:** 5

**Summary:**

In this work, the authors propose to train Language Models with multiple independent tokenizer and language modeling heads (one per language or data source) but with a single Transformer backbone. One touted advantage of this approach is that each source can train on its own GPU node, with a reduced need for cross-node communications (since a significant share of the weights does not need to be replicated across them). Another touted advantage is that forcing one model to work with multiple tokenization schemes makes the model more adaptable to new languages and domains post-training, similar to how "embedding reinitialization during the training" was found to have a similar effect (at a cost that however did not warrant its use when training large models, unlike the proposed methodology that does not requiring wasting training cycles).

**Strengths:**

- The setup proposed in this paper looks very satisfying, and it seems to solve several problems both in the industry and in research labs.
- The value proposition seems clear to me.
- The deployed methodology appears novel.
- The literature research looks satisfactory to me, given the scope of the paper.

**Weaknesses:**

- [addressed] The paper's form is well below the required writing standards. To address this, I'd suggest specific improvements, such as:
  - Standardizing method names throughout the paper and tables (SPED vs SPEC, GlOB vs GLOB vs Glob, ...)
  - Clearly defining the performance metrics used and specifying explicitly whether lower or higher values are better
  - Adding a reference to Table 1 in the main text
  - Improving table readability by adding summary statistics (averages...), using bold or color highlighting, or splitting into multiple tables (moving some languages to an Appendix).
- [addressed] Not enough arguments are brought forward to justify the issue with diverging models during training. I have myself never experienced this phenomenon in similar setups. As such, it's difficult to rule out that it might be the result of bugs in the training code or poor hyperparameter choices, rather a general phenomenon.
  - A better description of the exact training methodology and the hyperparameter search would help alleviate concerns, here.
  - An ablation study or an explanatory paragraph isolating factors that contribute to divergence would also help.
- [addressed] The lack of comparisons with baselines not trained by the authors is worrying.
  - I would prefer for external baselines to be added, even if some added context is necessary to explain away unfair comparisons (could be an appendix).
- [addressed] Without devising a clear methodology to perform inference on SPEC-type models, the paper feels a bit incomplete.
  - I'd suggest to the authors to briefly outline a proposed inference methodology for SPEC models, and to discuss the challenges and potential approaches for inference with these models in more detail.

**Questions:**

- [addressed] Is there any comparable experiment whose results you can compare your models against?
- [addressed] How did you perform hyperparameter search?
- [addressed] Did you perform scaling experiments?

[addressed] More than answering questions, I expect the authors to do a significant effort during the rebuttal to improve the paper's form. I would downgrade my rating if this does not happen. (I prefer to give authors a chance to amend, because I think the paper is otherwise interesting if you can get past the shortcomings of the writing.)

---

> ### Author Response · Authors · 2024-11-16
> **Response 1/2 to Reviewer vuvz**
>
> # Dear Reviewer **vuvz**,
>
> We genuinely appreciate the Reviewer's **thorough assessment** of our submission, which recognizes the **soundness**, **novelty**, and **careful literature review** shaping the foundation of our proposal. Their review is very insightful and has improved our submission. Below we elaborate on the following comments:
>
> ---
>
> - **RE: “Writing standard”**
>   We greatly appreciate the reviewer’s valuable suggestions. In the revised version of the paper:
>   - We have fixed and applied all the suggestions.
>   - Method names are made consistent, and performance metrics are clearly named in the tables.
>   - Arrows indicating whether greater or lower values are better have been added.
>   - All tables are now referenced in the main text.
>   - Tables report the average value as a separate column, highlight the best performance in bold, and include the improvement (%), as described in the captions.
>
> ---
>
> - **RE: “Model divergence”**
>   Thank you for raising this important point. The phenomenon of model divergence during training has been documented in several works (e.g., [1, 2, 4]), with recent theoretical insights provided in [6]. Divergence is often preceded by activation instability and can occur even in smaller transformers under specific conditions, such as:
>   - Unsuitably high learning rates [4, 6]
>   - Data heterogeneity [1]
>
>   **Widely accepted solutions to divergence include**:
>   - Extensive hyperparameter tuning
>   - Gradient clipping
>   - Expensive techniques like using “a token-distribution Kullback-Leibler divergence to filter out documents containing excessive numbers of outlier tokens compared to the training corpus distribution” [1].
>
>   DEPT mitigates this issue significantly, offering a **more robust and cost-effective framework** for model development.
>
>   **To address the Reviewer’s concern**:
>   - We have expanded **Section 3.4** to discuss contributing factors to divergence and their mitigations.
>   - All these explanations are added as per the reviewer’s suggestion.
>
>   **Code-related bug concerns**:
>   - Our training pipeline is built on the rigorously tested **MosaicML Composer library** for LLM pre-training and the **Flower framework** for federated learning.
>   - For our **InnerOPT implementation**, we rely on MosaicML's hyperparameters and infrastructure, modifying only the embedding matrix as per **Algorithm 1**.
>   - Standard baselines are trained using the unmodified MosaicML codebase, which is validated by wide adoption and peer-reviewed publications [7].
>   - Our codebase is available as an **anonymous GitHub repo** listed in **A.1.1**.
>
>   Additional clarifications about the training setup and hyperparameter choices have been incorporated into **Sections A.1, A.1.1, and A.1.2** of the revision.
>
> ---
>
> - **RE: “Hyperparameter search”**
>   We emphasize the following:
>   - For robustness plots, models are compared with the **exact same learning rate**.
>   - For performance comparisons, we always use **well-tuned checkpoints**, as reported in **Table 8** and the new **A.1.2** section.
>
>   **Details of hyperparameter tuning**:
>   - MosaicML provides hyperparameter-tuned models on the **C4 dataset**, and we use their learning rate schedule and number of training steps as a starting point.
>   - For DEPT, we find that MosaicML parameters work well since the **OuterOpt** acts as a regularizer.
>   - For standard training baselines, the learning rate is gradually lowered, starting from the one recommended by MosaicML (as shown in **Table 8**).
>     - Learning rates are reduced progressively by $5 \times 10^{-5}$.
>     - Starting learning rates range from $6 \times 10^{-4}$ to $2 \times 10^{-4}$, with larger models using smaller learning rates.
>
>   **Cosine cycle and validation perplexity**:
>   - We stop when we find a learning rate allowing us to complete a full cosine cycle for the pre-determined compute-optimal number of training tokens [3] (approximately 20 tokens per parameter).
>   - The best-performing checkpoint is chosen based on **validation perplexity**, across all experiments (**reported in Table 8**).
>
>   Gradient clipping is always applied with a **1.0 norm threshold**.
>
>   While this hyperparameter search does not cover all possible parameters (e.g., gradient clip norm, batch size), we are happy to incorporate any suggestions.
>
> ---
>
> - **RE: “External baselines”**
>   Thanks for this excellent suggestion.
>   - Results will be made available in the appendix with sufficient context for interpretation as soon as experiments finish.
>
>   **Key points**:
>   - Our goal is to propose a **novel training paradigm**, not to surpass the SOTA on downstream evaluation.
>   - Unlike SOTA small models, our models have not been trained on hundreds of billions of tokens from carefully curated data sources.
>
> ---

---

> > ### Author Response · Authors · 2024-11-16
> > **Response 2/2 to Reviewer vuvz**
> >
> > - **RE: “Using SPEC for inference”**
> >
> >     We thank the reviewer for this recommendation, which is a **great addition** to our work and to the practicality of DEPT’s models.
> >
> >     **Latest revision updates**:
> >
> >     - Additions have been made in **Sec. 2.4** (previously **Sec. 2.3**) and **Sec. 3.5**.
> >     - A new section, **Sec. F**, directly addresses this concern.
> >
> >     **Inference Methodology (See Sec 3.5)**:
> >
> >     - The work primarily evaluated transformer body performance and only implicitly proposed an inference method. This was in the form of simple continued pre-training in a **multi-phase adaptive pre-training framework** [8] on a broad non-private corpus starting from randomly initialized embeddings.
> >     - While this method requires additional training, more efficient methodologies may exist [9, 10].
> >
> > ---
> >
> > - **RE: “Comparable experiments”**
> > See **RE: “External baselines”**.
> >
> > ---
> >
> > - **RE: “How did you perform hyperparameter search?”**
> > See **RE: “Hyperparameter search”**.
> >
> > ---
> >
> > - **RE: “Scaling experiments”**
> >
> >     We have performed **scaling experiments** in the model size dimension, training two model sizes for both **The Pile** and **MC4**, as reported in the new **Section B.7** and **Fig. 4**.
> >
> >     **Key results**:
> >
> >     - DEPT performs better for larger models, likely due to **reduced parameter interference** during **OuterOpt**.
> >     - This is consistent with prior work on federated pre-training [11, 12].
> >     - Larger models may benefit because the transformer body has greater capacity for abstract language representations.
> >
> >     If the Reviewer believes additional scaling experiments would significantly improve the paper, we are happy to perform them.
> >
> >
> > ---
> >
> > As the Reviewer may have noticed, we are firmly committed to improving our submission thanks to the crucial comments and suggestions we received.
> >
> > In this first stage of the discussion:
> >
> > - We have uploaded a new revision addressing most of the Reviewer's comments.
> > - We are completing the additional evaluations requested.
> >
> > We hope the Reviewer maintains or increases their scores, especially in light of the **significantly improved presentation**, **clarifications**, and **experimental design**.
> >
> > ---
> >
> > ### References
> >
> > [1] Dubey et al., *The llama 3 herd of models*.
> >
> > [2] Zhang et al., *OPT: Open pre-trained transformer language models*.
> >
> > [3] Hoffmann et al., *Training compute-optimal large language models*.
> >
> > [4] Wortsman et al., *Small-scale proxies for large-scale transformer training instabilities*.
> >
> > [5] McCandlish et al., *An empirical model of large-batch training*.
> >
> > [6] Takase et al., *Spike No More: Stabilizing the Pre-training of Large Language Models*.
> >
> > [7] Blakeney et al., *Does your data spark joy? Performance gains from domain upsampling at the end of training*.
> >
> > [8] Gururangan et al., *Don’t stop pretraining: Adapt language models to domains and tasks*.
> >
> > [9] Remy et al., *Trans-tokenization and cross-lingual vocabulary transfers: Language adaptation of LLMs for low-resource NLP*.
> >
> > [10] Xu et al., *Bridging the gap between different vocabularies for LLM ensemble*.
> >
> > [11] Douillard et al., *Diloco: Distributed low-communication training of language models*.
> >
> > [12] Sani et al., *The future of large language model pre-training is federated*.
> >
> > **Kind regards,**
> >
> > *The Authors*

---

> > > ### Comment · Reviewer_vuvz · 2024-11-22
> > > **Thank you for your update**
> > >
> > > Your detailled answer was a pleasure to read, and I'm very happy to increase my Paper Presentation score based on your edits, as well as confirm my acceptance score. Congrats on your excellent work!

---

> > > > ### Author Response · Authors · 2024-11-22
> > > >
> > > > Dear Reviewer vuvz,
> > > >
> > > > We are joyous that our detailed comments addressed your initial concerns and that our work and rebuttal have matched your expectations, leading to an increase in the presentation score and a reaffirmation of the general score. We sincerely appreciate your fair and constructive critique from the beginning, which has been invaluable in guiding our approach throughout the rebuttal process.

---

> > > > > ### Comment · Reviewer_vuvz · 2024-11-26
> > > > >
> > > > > In light of the addition of additonial baseline experiments in the appendix, as well as new downstream tasks in the paper, I'm increasing my contribution score to Excellent as well. I think the presentation of the paper is fine; while it could still be improved a bit for the camera-ready version (e.g. some text elements are rather small), this shouldn't stop this excellent work from being presented at the conference. I think the ratings received by this paper are strong enough to ensure that at this point.

---

### Official Review · Reviewer_GATh · 2024-11-09

**Soundness:** 3
**Presentation:** 2
**Contribution:** 3
**Rating:** 8
**Confidence:** 4

**Summary:**

The authors introduce the method DEPT, “Decoupled Embeddings for Pre-trained Language models,” an algorithm that trains a transformer model on heterogeneous datasets such as domains or languages individually and subsequently aggregates the parameters by averaging them. They introduce three variants where increasingly less embedding information is shared. DEPT achieves lower perplexity than baselines.

**Strengths:**

1. The authors tackle an important problem. The usage of data mixtures during pre-training is not well understood but is an essential part of modern foundation models.
2. While the idea of using model averaging after an inner loop of training on dedicated subsets of data is not particularly novel, it might have a big impact on pre-training, given the encouraging results.

**Weaknesses:**

1. Writing can be improved or misses important information. For example, for the experimental setup, I struggle to understand L205-311, and information on software/hardware, such as how many FLOPS  or hours training took, is missing.
2. Some claims are overstated: M-T outperforms DEPT in 5/11 datasets in Table 2. I am not convinced that Trim and Glob perform identically (L377).
3. An important additional baseline would be models trained on individual data sets. This would give insights into the advantages/disadvantages of model averaging.

**Questions:**

1. How would the model DEPT-SPEC be used for inference? Would there first be a need to determine the used domain / language as tokenizers are separate (L129)?
2. The GLOB model seems to work best. Is there any hypothesis why this is the case? What is the implication of this fact?
3. The mentioned “Performance metrics” in all captions is always perplexity?
4. Based on which estimation is communication cost 675x lower?

---

> ### Author Response · Authors · 2024-11-16
> **Response 1/2 to Reviewer GATh**
>
> #
>
> # Dear Reviewer GATh,
>
> We thank the Reviewer for their valuable feedback and for acknowledging that our submission tackles an **important** problem and shows **promising** results. We now address the rest of the Reviewer’s comments and requests:
>
> ---
>
> - The Reviewer raises an excellent point regarding the origins of DEPT in parameter averaging methods. Despite its origins, DEPT significantly extends past traditional FL and parameter averaging methods, where identical models with the same objective function are trained in parallel and then aggregated:
>     - **TRIM** modifies the optimization objective by reducing the vocabulary to each data source’s specific data:
>         - **Loss Domain**: The cross-entropy loss is computed over a restricted token embedding space derived from the TRIM-specific vocabulary.
>             - Probabilities for a token at time-step $x_t$ are calculated as:
>                 - $p(x_t \mid x_{1:t-1}; \phi_{\text{TRIM}}) = \text{softmax}(\mathrm{score}(h_{t-1}, \phi_{\text{TRIM}}))$
>                 - Where $x_{1:t-1}$ is the sequence of previous tokens and  $\text{score} $ is a function which outputs a scalar summarizing the relation between the last hidden state $h_{t-1}$ and every token embedding in the $\phi_{\text{TRIM}}$ embedding matrix
>             - The cross-entropy loss on data source $k$ is:
>                 - $\mathcal{L}^k_{\text{TRIM}} = - \sum_{t=1}^n \log p(x_t \mid x_{1:t-1}; \phi_{\text{TRIM}})$
>     - **SPEC** extends heterogeneity further, allowing completely distinct tokenization schemes and vocabularies across clients:
>         - **Autoregressive Task**: Identical data may be tokenized into entirely different sequences (with varying lengths) based on each client’s unique vocabulary.
>         - **Loss Domain**: The cross-entropy loss is calculated over a data-source-specific set of tokens:
>             - $p(x_t^k \mid x_{1:t-1}^k; \phi_k) = \text{softmax}(\mathrm{score}(h_{t-1}, \phi_k))$
>             - The cross-entropy loss for data source $k$ is:
>                 - $\mathcal{L}^k_{\text{SPEC}} = - \sum_{t=1}^{n} \log p(x_t^k \mid x_{1:t-1}^k; \phi_k)$
>     - Thus, both variants **significantly depart** from standard methods for the scoring function, objective function, and, in the case of SPEC, the data representation itself.
>
> ---
>
> - **RE: Writing clarity and missed information**
>
>     We apologize for the confusion. An improved description of the use of baselines and metrics is available in **Sections 3.3** and **3.4** in the revised manuscript. The software, which we make available, is based on the **Flower framework** [1] and the **MosaicML Composer library** [2]. As for the hardware, we use a mixture of cloud-based instances equipped with a range of Nvidia GPUs, e.g., **H100** and **A100**, in several regions (e.g., Europe, US, and Canada). The details are in the revised version (**Sec. 3.3**, **3.4**, and **A.1.1**), we also make our FL pre-training framework available in **A.1.1** in an anonymized form.
>
>
> ---
>
> - **RE: Wall-clock training time**
>
>     While DEPT takes the same number of training steps as baselines, assuming the same batch size and number of GPUs, two factors influence time:
>
>     - **Worker throughput**: For GLOB, it matches standard pre-training as the model  is identical. For TRIM and SPEC, reduced memory requirements allow increasing micro-batch sizes in some hardware-dependant settings and hence faster training—e.g.,, doubling it on an **H100** for a **24-block model** for the small-vocabulary DeepMind Mathematics tasks.
>     - **Communication topology**: On cloud machines with **10 Gbps bandwidth** and **Ring AllReduce**, DEPT reduces training time by **33%** for a **1-billion parameter model**. On faster connections like **InfiniBand**, differences are primarily due to throughput.
>
> ---
>
> - **RE: M-T performance**
>
>     Insightful observation. We note that DEPT wins **82.2%** of comparisons overall and hope this figure sufficiently justifies our outperformance claims. For example:
>
>     - We win **all comparisons** on downstream tasks and a vast majority of the language modeling comparisons.
>     - **Table 2**, in the original submission, shows results starting from a pre-trained embedding matrix, which are sensitive to the sampling used by the baselines during pre-training. Looking at the updated version of **Table 2** (**Table 6** in the latest revision), we see that DEPT embeddings are slightly worse for high- and medium-resource languages like EN, IT, or SR, or for LA (closely related to EN and IT). However, DEPT wins the comparisons for very low-resource UR and SW languages by large margins.
>     - Furthermore, in the case of **The Pile** (**Table 5** in the new version), DEPT still wins **12/17** comparisons under this exact setting, showing that it is not a permanent disadvantage.
>     - Since embeddings are much easier to transfer or retrain, we argue that obtaining a better transformer body is generally preferable.

---

> > ### Author Response · Authors · 2024-11-16
> > **Response 2/2 to Reviewer GATh**
> >
> > ---
> >
> > - **RE: TRIM and GLOB not being identical**
> >
> >     In the revised manuscript, we have made it clear that these two variants are close in terms of validation perplexity but not identical.
> >
> >
> > ---
> >
> > - **RE: Additional baseline**
> >
> >     Thanks for introducing this interesting baseline. These experiments are currently being executed. Given that training the **24-block model** takes upwards of 24 hours with our resources, we hope to add at least **four** such models for each dataset—two for high-resource subsets and two for low-resource subsets. The newly added unigram cross-entropy provides a proxy indicating how difficult each subset is for the tokenizer.
> >
> >
> > ---
> >
> > - **RE: Inference with DEPT-SPEC**
> >
> >     We use continued pre-training on a broad corpus to build a general embedding matrix as a form of multi-phase adaptive pre-training [3]. You also need to know the vocabulary of the target inference data source, which may depend on the domain or language, to obtain good embeddings for it. If you have multiple pre-trained tokenizers, as SPEC produces, you can select which one to use based on the cross-entropy of the unigram model it defines over the target dataset.
> >
> >
> > ---
> >
> > - **RE: GLOB works best**
> >
> >     Insightful observation. While we agree that GLOB performs well in many scenarios, it does not emerge as an undisputed winner:
> >
> >     - In our new experiments, including the evaluations on downstream tasks (**Table 7**), TRIM outperforms GLOB in **five out of six** comparisons.
> >     - SPEC often outperforms GLOB on **OOD data**.
> >     - In the situations where GLOB does perform well for language modeling, we hypothesize that increased parameter sharing and potential memorization (given the outperformance of the other methods on OOD data or downstream tasks) plays a role. When GLOB is usable for the non-private deployment setting and if it outperforms the other variants, the implication is that adopting a shared embedding layer would be preferable for this restricted case.
> >
> > ---
> >
> > - **RE: Performance metrics**
> >
> >     The performance metric in those tables is perplexity, and we have amended the work. Additionally, we have performed new downstream evaluations that use other metrics in **Table 7**.
> >
> >
> > ---
> >
> > - **RE: 675x communication improvements**
> >
> >     Our estimation for this significant improvement is based on two elements:
> >
> >     - First, unlike standard data-parallel mini-batch SGD, where gradients are communicated every step, DEPT communicates models between GPU workers every **N=500** training steps. Since we use **N=500**, we obtain a **500x** reduction with GLOB.
> >     - Second, on top of the above, SPEC completely eliminates the need to communicate the embedding parameters across workers. In the case of the billion-scale multilingual model, these represent **30%** of parameters. This results in a total reduction of **714x**, which replaces the previously miscalculated **675x**.
> >     - We acknowledge that this explanation was lacking in the paper. In the revised version, we have added the justifications. The asymptotic complexity of all methods is now shown in **Table 1**, and the practical communication and memory costs of our models are shown in **Table 2** and **Table 9**.
> >
> > ---
> >
> > We are firmly committed to improving our submission thanks to the crucial comments and suggestions we received. In this first stage of the discussion:
> >
> > - We have uploaded a new revision addressing most of the Reviewer's comments.
> > - We are completing the additional evaluations we have been asked to produce.
> >
> > We hope to improve our work progressively, and we kindly ask the Reviewer to raise their scores if they believe our rebuttal addresses their concerns in light of our **additional experiments**, **clarifications**, and **significantly improved presentation**.
> >
> > ### References
> >
> > [1] Beutel et al., *Flower: A friendly federated learning research framework*.
> >
> > [2] Composer. https://github.com/mosaicml/composer
> >
> > [3] Gururangan et al., *Don’t stop pretraining: Adapt language models to domains and tasks*.
> >
> > **Kind regards,**
> >
> > *The Authors*

---

> > > ### Comment · Reviewer_GATh · 2024-11-22
> > >
> > > Thank you for your detailed response! My feedback and questions were addressed appropriately and the updated version of the paper improved the presentation. Thus, I increase the score to 8 and support the acceptance of the paper.

---

> > > > ### Author Response · Authors · 2024-11-22
> > > >
> > > > Dear Reviewer GATh,
> > > >
> > > > Thank you very much for your inquisitive feedback which has been highly beneficial to our work and for increasing your score in light of our improved evaluation and presentation.

---

### Author Response · Authors · 2024-11-16
**General Rebuttal**

# Dear Reviewers, ACs, and PCs,

Thank you very much for your dedication, support, and insightful feedback. We are delighted you have found our work and experimental design **insightful** and **sound** [vuvz], addressing an **important** [GATh] and **novel** [xZ5j, vuvz] problem by decoupling the embedding matrix and transformer body. Your constructive comments have immensely helped us improve our paper's quality. We have reviewed all the comments, addressed all questions, and provided new experimental results. Below, we summarize the revisions we made:

---

## Revisions and Updates

### Experimental Results

- **Additional experimental results. A total of 38 new experiments added:**
    - [GATh] DEPT variants win **82.2% of experiments** in terms of perplexity and performance on downstream tasks.
    - [xZ5j] Fine-tuned DEPT models on tasks like **Natural Language Inference**, **Question Answering**, and **Sentence Similarity**, outperforming baselines in all configurations (**Table 7**).
    - [GATh] Studied sampling ratio choices during pre-training on **The Pile** dataset:
        - Switching from uniform to proportional sampling improved results (**Tables 3** and **5**).
    - [GATh, vuvz] Initiated **26 additional experiments** (results forthcoming):
        - Baselines where local models are trained on individual data sources.
        - Comparisons with externally trained models.

### Writing Standards and Clarity

- [GATh, vuvz, xZ5j] Improved clarity of methods and experiments by providing detailed explanations and further context.

### Presentation of Results

- Enhanced **tables**, **figures**, **captions**, and **descriptions** to improve readability and presentation consistency. Perplexity values too large to fit in tables are now denoted by a “-“.

### Updated Workflow Illustration

- Modified **Figure 1** to better showcase DEPT’s workflow and compare its variants to standard pre-training pipelines.

---

## Section-by-Section Walkthrough of Revisions

**Revisions are highlighted in blue in the manuscript.** Key changes include:

- **Abstract:**
    - Added figures illustrating DEPT’s improved perplexity and downstream fine-tuning performance.

- **Section 1:**
    - Provided numerical results for DEPT’s **generalization performance**.
    - Detailed the **memory footprint** and **communication efficiency** in asymptotic terms.

- **New Section 2.2:**
    - Formally described the **TRIM variant trimming procedure**.

- **Section 2.4 (previously Section 2.3):**
    - Clarified inference steps for **SPEC-trained models**, with further details in **Section 3.5** and **Appendix F**.

- **New Table 1 (Section 2):**
    - Summarized **asymptotic complexities** for memory and communication costs.

- [GATh] **New Table 2 (Section 4):**
    - Addressed scalability and communication efficiency (**714x improvement**):
        - **500x reduction** from infrequent averaging.
        - **30% reduction** from excluding embedding parameter communication.

- **Appendix A.1:**
    - Detailed **hyperparameter choices** and **sampling strategies** for continued pre-training.

- [GATh] **Appendix A.1.1:**
    - Outlined **software/hardware configurations** and provided an anonymous GitHub repository with DEPT execution code.
    - Reported hardware throughput and time differences between DEPT and baseline methods.

- [vuvz] **Appendix A.1.2:**
    - Detailed the **hyperparameter tuning methodology** for both DEPT and baselines.

- **Appendix A.1.3:**
    - Adapted active forgetting methodology for decoder-only models.

- **Appendix A.2:**
    - Additional statistics on data source heterogeneity, including unique token counts.

- **Appendix A.2.1:**
    - Described the unigram normalized cross-entropy, presented in **Tables 3–6**, as a reference metric for data source difficulty.

- **[GATh, vuvz, xZ5j] Appendix B:**
    - Presented results on:
        - **Larger models** (**B.1**).
        - **Model plasticity** (**B.2**).
        - **IID data performance** (**B.3**).
        - **Generalization** (**B.4**).
        - **Scaling from smaller models** (**B.7**).

- [xZ5j] **Appendix E:**
    - Added fine-tuning parameter details.

- [GATh, vuvz] **Appendix F:**
    - Presented multiple inference methods for SPEC.

---

## Request for Feedback

We kindly ask the reviewers to inspect our revisions and individual responses. We are happy to address any remaining questions. If our responses and additional results adequately address your concerns, we kindly ask you to consider increasing your scores. We deeply appreciate your engagement and suggestions, which have been invaluable for improving this work.

**Best regards,**

*The Authors*

---

> ### Author Response · Authors · 2024-11-26
>
> # Dear Reviewers, ACs, and PCs,
>
> We have completed another paper revision, addressing all reviewers' requests and providing the promised baselines as detailed in the general comments above.
>
> ### 1. **Impact of Model Averaging (GATh's Recommendation)**
>
> To evaluate the impact of model averaging, we compared models undergoing continued pre-training from a DEPT initialization with models undergoing continued pre-training starting from models trained on isolated data sources. For a fair comparison, each isolated model sees as many tokens as its DEPT-based counterpart and undergoes continued pre-training for the same number of steps, with access to the full dataset. The results of this comparison are detailed in Section B.8, they show that the model averaging allows DEPT models to generalize better across distributions.
>
> ### **Continued Pre-Training from Random Initialization**
>
> - This comparison tests the generalizability of the abstractions learned by the transformer body.
>     - **[Table 14] The Pile**: DEPT-based models outperform the baselines by **7.8%** in average perplexity.
>     - **[Table 15] MC4**: DEPT-based models outperform baselines by **9.8%** on in-distribution data and **16.7%** on out-of-distribution (OOD) data.
> - Despite isolated models receiving embeddings suitable for the multi-dataset context through the continued pre-training process, their transformer body abstractions do not generalize well across distributions.
>
> ### **Continued Pre-Training from Pre-Trained Embeddings**
>
> - Models initialized from isolated data-source models are biased toward their pre-training data, performing well on specific datasets but failing to generalize:
>     - **[Table 16] The Pile**: DEPT-based models outperform baselines by a remarkable **27%** in average perplexity. However, DEPT models tend to be outperformed by isolated baselines for their respective data sources.
>     - **[Table 17] MC4**: DEPT-based models achieve significant outperformance, with a **30.6%** improvement in average perplexity for in-distribution data and **14.9%** for OOD data. This is likely because embeddings generalize worse across languages than across domains.
>
> ---
>
> ### 2. **Comparison Against Externally-Trained Baseline (vuvz's Recommendation)**
>
> We compared our models with the externally trained Pythia[1] model suite, which uses a similar architecture but untied weights for the embedding matrix. We chose Pythia as the closest baseline in terms of model structure and pre-training dataset. As a result of the untied embedding weights, Pythia models with the same number of blocks have more parameters. Pythia models are trained on one epoch of The Pile (300B tokens compared to 10B to 30B for DEPT models), thus, they are expected to perform better on Ubuntu IRC than DEPT models since it is not OOD for Pythia. These comparisons have been added to section B.9, they show that DEPT is competitive with Pythia at smaller scales and when starting continued pre-training from random initialization (indicating a comparable transformer body); however, the superior embeddings of the larger Pythia model, pre-trained for 10x more tokens, allow it to outperform DEPT models when using pre-trained embeddings.
>
> ### **160M Model Comparison**
>
> - **[Table 18]** For **Pythia-160M** vs. **DEPT-125M**:
>     - DEPT models (with pre-trained embeddings) outperform Pythia by **1 point** in average perplexity.
>     - Pythia's extensive pre-training (30x DEPT's token budget) does not provide an evident advantage due to insufficient model capacity to benefit fully from the additional data.
> - Random initialization results for this model size are omitted as Pythia models were unstable during the continued pre-training, making the comparison unfair.
>
> ### **410M Model Comparison**
>
> - For **Pythia-410M**:
>     - **[Table 19] Random Initialization**: DEPT's best variant is within **1 point** of Pythia-410M in average perplexity, suggesting that much of Pythia's additional token budget enhances embeddings without significantly improving the transformer body.
>     - **[Table 20] Pre-Trained Embeddings**: Pythia-410M significantly outperforms DEPT, achieving **10 points lower** in average perplexity, likely due to training on 10x more tokens than the equivalent DEPT models and thus achieving superior embeddings.
>
> ---
>
> ### 3. **Addition of a New Downstream Task (xZ5j's Recommendation)**
>
> We added a new downstream task, **Sentence Classification** evaluated on the SST-2 dataset[2], and reported the results in Table 7. DEPT models outperform all baselines on this task, achieving a **3.2% to 4.1%** relative improvement in accuracy.
>
> These experiments have helped significantly improve our work, and we thank the reviewers again for recommending them.
>
> **Kind Regards**,
>
> *The Authors*
>
> [1] Biderman, et al., “Pythia: A Suite for Analyzing Large Language Models
> Across Training and Scaling”
>
> [2] Socher, et al., Recursive deep models for semantic compositionality over a sentiment treebank

---

### Author Response · Authors · 2024-11-26
**Rebuttal Reflections**

# Dear Reviewers, ACs, and PCs,

We sincerely thank the reviewers for their time, constructive feedback, and engagement throughout the discussion phase. Your insights have been invaluable in improving the clarity, presentation, and depth of our submission. We are thrilled to hear that our revisions and additional experiments addressed your concerns, and we deeply appreciate the positive updates to your evaluations.

### **Discussion Highlights**

- Reviewer **GATh** initially recognized our work as **tackling an important and novel problem** while providing a **method that could significantly impact pre-training methodologies**. After reviewing our detailed responses, improved presentation, and additional experiments, the Reviewer updated their score to 8, highlighting that **our updates adequately addressed their concerns** and endorsed the acceptance of our paper with **high** confidence.
- Reviewer **vuvz** praised the **clear value proposition and novelty** in their initial feedback. After seeing our improvements in presentation, hyperparameter search explanations, and inference methods, they affirmed their support, stating, **“Your detailed answer was a pleasure to read”,** and maintained their score of 8 with firm support for acceptance with **very-high** confidence.
- Reviewer **xZ5j** commended the paper for being **well-written, easy to follow, and methodologically novel**. Our response, particularly the downstream task evaluations, and clarifications, led them to declare our work as **“a good paper worth accepting”** and increase their score to 8  with **high** confidence.

### A complete set of new experimental results

Throughout this rebuttal, we have provided a new set of experimental results based on the reviewers' feedback. The first set accompanied revision 1 of the paper and included 26 additional experiments covering downstream tasks[**xZ5j**] and continued pre-training sampling policies. We provide the second set of results now in revision 2, covering 51 new experiments, including comparisons against the externally-trained Pythia 160M and 410M baselines[**vuvz**], an additional downstream task[**xZ5j**], and comparisons against new baselines trained on isolated data sources without model averaging[**GATh**].  For details on both sets of experiments, please look at our previous [general rebuttal comment](https://openreview.net/forum?id=vf5aUZT0Fz&noteId=d7Krp7Zo3V) and its [reply](https://openreview.net/forum?id=vf5aUZT0Fz&noteId=GLNSHb0pot).

### Impact of DEPT

We envision DEPT as the foundation of a **new paradigm for pre-training language models**. This framework **uses decoupled pre-training on heterogeneous data** to obtain a more generalizable and adaptive/plastic transformer body than the one produced by standard pre-training methods.

The implications of this paradigm are:

- **Flexibility:** Practitioners can train on heterogeneous and even private data sources without the complexities of creating a shared vocabulary.
- **Efficiency:** The reduced communication and memory costs allow pre-training across geographically distributed GPUs, enabling a wider-scale deployment than standard pre-training.
- **Generality:** DEPT-trained models can serve as versatile foundation models that perform well across various tasks and domains. Compared to standard pre-training, this improved foundation model can enhance the efficacy of continued training and fine-tuning.

This approach not only simplifies training pipelines but also extends the applicability of pre-trained models to settings where diverse, distributed, and sensitive datasets must be leveraged.

### Conclusion

We are delighted that the reviewers now unanimously consider our paper to be **worth accepting** and appreciate the value and novelty of DEPT. We hope DEPT’s contributions and vision resonate with the community as a step forward in multilingual and multi-domain model pre-training.

Thank you again for your support, organization, and invaluable feedback.

Best regards,

The Authors

---

### Meta-Review · Area_Chair_kdZ4 · 2024-12-06

**Metareview:**

This work proposes the DEPT algorithm (Decoupled Embeddings for Pre-Trained language models) to train LMs on diverse data sources more effectively. The method iteratively trains a transformer model on different subsets of data with separate embeddings for each data type (such as domain or language); the work proposes and compares three variants of this approach. The experiments show that this approach performs well for continued pretraining in both different English domains and in the multilingual setting, and the analyses also demonstrate the method's efficiency.

Strengths:
- This paper addresses an important problem (tokenization and embedding representations over diverse data sources) and provides a compelling approach to solving this problem.
- Results are (for the most part) consistent and convincing for both the perplexity and downstream task experiments.
- The approach is more efficient than standard pretraining, as cross-node communication isn't needed during the steps where the model is trained independently on each data type (InnerOPT).
- The paper provides clear research goals beyond task performance that the experiments and analyses address, such as model efficiency and generalization.


Weaknesses:
- Multiple reviewers raised concerns with the writing, and some details are still confusing after revisions (e.g., defining STANDARD as the continued pretraining baseline in Table 2 and then referring to it with the names based on sampling methods in Tables 3 and 4). The authors should ensure the notation and discussion are consistent and clear in the next version of the paper. This also includes other points raised by the reviewers and clarified in the rebuttal.
- Table 7 is too small to read easily and should be larger in the next version.

**Additional Comments On Reviewer Discussion:**

The authors provided detailed responses and a revised draft based on the reviews, with a main focus on clarifying points that were not clear in the original text and adding downstream task evaluations. Reviewers who did not already recommend acceptance (a score of 8) increased their score after the discussion period.

---

### Decision · Program_Chairs · 2025-01-22

Accept (Oral)